# 30 years of total column ozone and aerosol optical depth measurements using the Brewer spectrophotometer in Poprad-Gánovce, Slovakia

Peter Hrabčák<sup>1</sup>, Meritxell Garcia-Suñer<sup>2</sup>, Violeta Matos<sup>2</sup>, Víctor Estellés<sup>2</sup>, Anna Pribullová<sup>1</sup>, Jozef Depta<sup>1</sup>, Martin Staněk<sup>3</sup>, and Martin Stráník<sup>3</sup>

Correspondence to: Peter Hrabčák (peter.hrabcak@shmu.sk)

Abstract. Long-term observations (1993–2024) of total column ozone (TCO) and aerosol optical depth at 320 nm (AOD<sub>320</sub>) from the Brewer spectrophotometer at Poprad-Gánovce, Slovakia, were analysed. The data used in this study are closely tied to three environmental issues driven by human activities: ozone depletion, climate change, and air pollution caused by aerosols. Both parameters exhibit distinct seasonal variability, with AOD<sub>320</sub> peaks in April and August and TCO maxima from February to April. AOD<sub>320</sub> shows a statistically significant long-term decrease ( $-0.057 \pm 0.005$  per decade), reflecting reduced anthropogenic emissions. The lowest annual mean occurred in 2020, corresponding to the first year of the COVID-19 pandemic. In contrast, TCO shows no clear trend ( $0.0 \pm 0.4$  DU per decade), whereas the tropopause height increases markedly ( $105 \pm 7$  m per decade) and exerts an inverse influence on TCO. After removing this effect, a positive TCO trend ( $1.3 \pm 0.3$  DU per decade) appears, while the tropopause-related component decreases ( $-1.3 \pm 0.1$  DU per decade). The most significant positive linear trend in tropopause height was identified in August ( $200 \pm 70$  m per decade), while the strongest negative TCO trend also occurred in August ( $-3.6 \pm 1.8$  DU per decade). Only about one third of this August TCO decrease is explained by the tropopause rise. LOTUS regression confirmed the inverse TCO-tropopause relationship and revealed significant ENSO (positive) and QBOB (negative) influences. A weak positive trend in TCO since 1997 ( $2.0 \pm 1.2$  DU) indicates ongoing recovery from ODS reductions.

#### 1 Introduction

Ozone depletion is caused by human-related emissions of ozone-depleting substances (ODSs) and the subsequent release of reactive halogen gases, particularly chlorine and bromine, in the stratosphere (WMO, 2022). Monitoring long-term changes in ozone, a vital component of Earth's atmosphere, is essential for assessing the effectiveness of the Montreal Protocol (UNEP, 2020). This protocol was established to regulate and significantly reduce the levels of persistent ODSs. Over the past two decades, measures implemented under the Montreal Protocol and its amendments have resulted in a substantial decline in the total concentration of ODSs (Braesicke et al., 2018). In addition to ODSs, model simulations indicate that

<sup>&</sup>lt;sup>1</sup>Aerological and Solar Radiation Center, Slovak Hydrometeorological Institute, Gánovce, 058 01, Slovakia

<sup>&</sup>lt;sup>2</sup>Department of Earth Physics and Thermodynamics, Universitat de València, Valencia, 46100, Spain

<sup>&</sup>lt;sup>3</sup>Solar and Ozone Observatory, Czech Hydrometeorological Institute, Hradec Králové, 500 08, Czech Republic

stratospheric ozone concentrations are also influenced by the chemical and climatic effects of greenhouse gases. Specifically, increasing concentrations of the greenhouse gases carbon dioxide (CO<sub>2</sub>) and methane (CH<sub>4</sub>) during this century are expected to raise global total column ozone (TCO) levels beyond the natural levels observed in the 1960s. This increase is primarily due to the cooling of the upper stratosphere and changes in stratospheric circulation. Conversely, the chemical effect of rising concentrations of nitrous oxide (N<sub>2</sub>O), another greenhouse gas, is to deplete stratospheric ozone (WMO, 2022).

TCO varies strongly with latitude over the globe, with the largest values occurring at middle and high latitudes during most of the year. This distribution is the result of the large-scale circulation of air in the stratosphere that slowly transports ozone rich air from high altitudes in the tropics, where ozone production from solar ultraviolet radiation is largest, toward the poles. Ozone accumulates at middle and high latitudes, increasing the vertical extent of the ozone layer and, at the same time, TCO.

The TCO is generally smallest in the tropics for all seasons. An exception since the mid-1980s is the region of low values of ozone over Antarctica during spring in the Southern Hemisphere, a phenomenon known as the Antarctic ozone hole (Salawitch et al., 2023).

TCO also varies with season. During spring, it exhibits maxima at latitudes poleward of about 45° N in the Northern Hemisphere and between 45° and 60° S in the Southern Hemisphere. These spring maxima are a result of increased transport of ozone from its source region in the tropics toward high latitudes during late autumn and winter. This poleward ozone transport is much weaker during the summer and early autumn periods and is weaker overall in the Southern Hemisphere (Salawitch et al., 2023). Furthermore, it has been reported that the Brewer-Dobson circulation seems to have accelerated during the last years due to the increased presence of greenhouse gases in the atmosphere (Braesicke et al., 2003; Butchart et al., 2006). Other natural atmospheric cycles (e.g., the Quasi Biennial Oscillation, El Niño-Southern Oscillation, Arctic and Antarctic Oscillations, the solar cycle, etc.) have also been found to influence TCO (Coldewey-Egbers et al., 2022). Since these cycles operate on different timescales, assessing the individual impact of each on TCO is challenging.

In the middle and high latitudes, the anticipated increase of TCO related to ODSs is obscured by significant dynamically induced interannual variability in ozone. Additionally, complex feedback mechanisms driven by climate change can result in further long-term variations in ozone levels, for instance, through changes in temperature (Coldewey-Egbers et al., 2022).

According to Coldewey-Egbers et al. (2022), during the period 1995–2020, the middle latitudes of the Northern Hemisphere exhibited distinct regional trends, characterized by both latitudinal and longitudinal structures. Significant positive trends (1.5 ± 1.0 % per decade) were observed over the North Atlantic region, while barely significant negative trends (-1.0 ± 1.0 % per decade) were identified over Eastern Europe. Furthermore, these trends correlate with long-term changes in tropopause pressure.

The trend in tropopause altitude has been positive over recent decades at a global level, primarily driven by thermal effects due to the warming of the troposphere as a consequence of increasing greenhouse gas concentrations (Santer et al., 2003; Pisoft et al., 2021). For the European station Hohenpeißenberg, data from 1967 to 1997 revealed that approximately 25 % of the negative long-term trend in TCO could be attributed to a tropopause that rose by 150 ± 70 meters per decade (Steinbrecht et al., 1998). Positive correlations with the trend in ozone were found for the northern Pacific, for both the southern and

northern parts of the North Atlantic, and for Europe according to Coldewey-Egbers et al. (2022). This study further states that the regions indicating a negative trend in ozone also show a negative trend in tropopause pressure with slopes between – 3.0 and –3.5 hPa per decade during the period 1995–2020. The negative pressure trend corresponds to an increase of about 100–130 m per decade in tropopause height. On the other hand, the northern part of the North Atlantic indicates a positive trend both in ozone (1.2 ± 1.2 % per decade) and in tropopause pressure. The latter is about 2 hPa per decade in that region (Coldewey-Egbers et al., 2022).

It is well known that anthropogenic changes in TCO levels and atmospheric aerosols significantly impact the solar UV radiation reaching the Earth's surface (Kim et al., 2013; De Bock et al., 2014). It was found that the overall mean radiation amplification factor (RAF) due to TCO, RAF(AOD, SZA) (Madronich, 1993), shows that 1 % decrease in TCO results in an increase of  $1.18 \pm 0.02$  % in the erythemal UV irradiance with the range of 0.67-1.74 % depending on solar zenith angles (SZAs) (40–70°) and on aerosol optical depths (AODs) (< 4.0), under both clear (cloud cover < 25 %) and all sky conditions. A similar analysis of the RAF(TCO, SZA) due to AOD under clear and all-sky conditions shows that on average, a 1 % increase in AOD forces a decrease of  $0.29 \pm 0.06$  % in the erythemal UV irradiance with the maximum range 0.18-0.63 % depending on SZAs and TCO (Kim et al., 2013).

The study by Filonchyk et al. (2020) provides characteristics of aerosol columnar properties based on MODIS-Aqua measurements over ten countries in Eastern Europe (including Slovakia) from 2002 to 2019. A gradual reduction in AOD (at 550 nm) is observed in all countries, with decadal trends ranging from -0.050 (Belarus) to -0.029 (Russia). Slovakia is characterized by a trend of -0.041 per decade. Results related to AOD at the Poprad-Gánovce station were previously published (Hrabčák, 2018). In that publication, data from 1994 to 2016 were analysed, representing a 23-year series of measurements. For the wavelength of 320 nm, the trend value was  $-0.05 \pm 0.01$  per decade. The following lines will update the information on the development of AOD at the Poprad-Gánovce location, as the series of measurements has already exceeded the 30-year milestone.

#### 2 Methods

# 2.1 Location

The Brewer spectrophotometer is located on the roof of the Aerological and Solar Radiation Center in Poprad-Gánovce, part of the Slovak Hydrometeorological Institute. Its coordinates are 49.03° N, 20.32° E (Central Europe), at an altitude of 709 meters above sea level. The site is situated in the Podtatranská Basin, which forms part of the larger Carpathian geomorphological unit. The surrounding area includes mountain ranges of varying elevations. Gerlachovský štít (2654 m above sea level), the highest peak in the Carpathians, is only 20 km from the station. Local aerosol sources primarily include emissions from burning solid fuels, mainly wood, in nearby villages, as well as agricultural activities. The area experiences relatively strong winds when influenced by a larger pressure gradient, with prevailing wind directions from the west, north, and southeast.

The proximity of the city of Poprad, located approximately 1.5 km away with a population of around 50 000 and various industrial activities, also influences the site. Despite its closeness to Poprad, the area is generally considered rural in terms of anthropogenic impact. Given that ozone is concentrated primarily in the stratosphere, local effects through ground-level ozone have only a minimal impact on its final value. The TCO above the measurement site at any time of the year depends almost exclusively on large-scale atmospheric circulation in the stratosphere.

#### 2.2 Brewer ozone spectrophotometer

The Brewer ozone spectrophotometer (a single monochromator, model MKIV, No. 97) has been performing measurements at the Poprad-Gánovce station since 18 August 1993. It is a scientific instrument that operates in the ultraviolet and visible regions of the solar spectrum. The device enables measurements of the total vertical column of O<sub>3</sub>, SO<sub>2</sub>, and NO<sub>2</sub>, as well as global UV radiation (from 290 nm to 325 nm, with a step of 0.5 nm). Using its optical system, the instrument decomposes solar radiation reaching the Earth's surface and selects predetermined wavelengths from the ultraviolet and visible parts of the spectrum, with stronger and weaker absorption by O<sub>3</sub>, SO<sub>2</sub>, and NO<sub>2</sub>. Based on the differing absorption of radiation at these selected wavelengths, it is possible to derive the total amount of these gases in the vertical column of the atmosphere. Additionally, measurements of direct solar radiation can be used to determine the AOD. The Brewer also enables the calculation of TCO using measurements of diffuse solar radiation. However, for the analysis of the TCO in this study, only direct solar radiation measurements were used, as they are more accurate.

Since the beginning of its operation, daily tests have been performed on the Brewer spectrophotometer using internal lamps, and it undergoes calibration every two years. The instrument is calibrated by International Ozone Services (IOS) Inc. against the global reference group (Brewer Triad). IOS is a long-established company that provides worldwide ozone and UV calibration services to customers operating Brewer Ozone Spectrophotometer instruments (https://www.io3.ca/; last access: September 2025). The calibration is carried out during a calibration campaign, usually on site or in a neighboring country, using a traveling reference instrument No. 17. The Brewer No. 17, used by IOS for calibrating Brewers belongs to Environment and Climate Change Canada. When not travelling, it is stationed next to the World Brewer Reference Triad in Toronto and is frequently compared with the triad instruments.

The calibration of Brewer No. 17 is usually updated annually, and more frequently if required, based on the data collected in Toronto. Over the years, IOS has also taken the instrument, together with the triad instruments, to the Mauna Loa Observatory (MLO) in Hawaii, USA, for independent calibrations. These trips to MLO and the regular comparisons with the triad have resulted in differences of no more than 0.5 % in TCO values between No. 17 and the triad. Further details on the calibration of Brewer #017 can be found in Savastiouk et al. (2004), and the Brewer reference triad is described in more detail in Fioletov et al. (2005).

The measurements can be regarded as homogeneous. Occasional short interruptions occurred for technical reasons, but no extended gaps such as entire months are present. Regarding major technical interventions or problems, these can be

summarised as follows: a secondary power supply board had to be replaced in January 2005. In February 2007, a micrometer was replaced, and during the calibration in May 2007, optical filter No. 3 was replaced and a BM-E80 high-frequency source was also repaired.

# 2.3 TCO calculation

155

- The TCO data set consists of daily averages collected by the Brewer ozone spectrophotometer. These data cover the period from 18 August 1993 to 31 May 2024. All data used were derived from direct sunlight measurements obtained through the DS (direct sun) measurement procedure. A DS measurement is accepted only for an air mass factor of the ozone layer of less than 4 (as recommended by IOS), and it takes approximately 2.5 minutes. During this time, the density of solar radiation flux is measured five times for each of the five wavelengths. Consequently, five values of TCO in Dobson units (DU) are obtained from a single DS measurement, which are then used to calculate an average and a standard deviation. Only the measurements that meet the criterion (standard deviation ≤ 2.5 DU) are selected for further data analysis. The TCO was calculated using the Brewer spectrophotometer B data files analysis program v. 7.4 (O3Brewer) by Martin Stanek (http://www.o3soft.eu/; last access: May 2025). Brewer spectrophotometer data file (B-file) contains the raw data collected by the Brewer spectrophotometer.
- The O3Brewer software employs initialization files that are typically updated during calibration and contain all necessary instrumental constants and settings. The TCO data set was derived using as many as 57 such files in total. This number is considerably higher than both the number of performed calibrations and the years of operation. The reason is that, when required (mostly due to major instrumental changes), the nominal two-year intercalibration interval was subdivided into shorter segments. Such finer partitioning was usually carried out retrospectively after the subsequent calibration. The key constants required for the calculation of TCO from DS measurements are determined and verified during calibration.
  - Among the most important are the ETC values and the absorption coefficients for O3. In connection with the standard lamp (SL) correction, the initialization file contains the so-called "values of SL R6 and R5 from the last intercomparison". The SL test O3 correction is performed such that the O3Brewer software begins reading the B-files 10 days prior to the selected time period in order to process the SL test results first and generate the "SLsmooth.prn" file, and continues until 10 days after the selected period. It is customary to perform the SL test three times per day. In addition, frequent mercury lamp tests are routinely carried out, usually when the internal temperature changes by more than 2 °C. Further information on the O3Brewer software can be found in the publicly available online documentation (http://www.o3soft.eu/o3brewer.pdf; last access: May 2025). In Appendix B, the presented TCO dataset is compared with existing observations archived in the World Ozone and Ultraviolet Radiation Data Centre (WOUDC) and the European Brewer Network (EUBREWNET).
- Finally, it should be noted that the correction for the stray-light effect has been applied during instrument calibrations only since 2023. This correction has also been taken into account in the calculation of TCO using the O3Brewer software, but only for data starting from 27 June 2023. However, past calibration procedures (before 2023) effectively compensated, at

least partially, for the stray-light effect of the instrument. For more information on this issue, further details can be found in Appendix C. A comparison between two Brewer spectrophotometers, the MKIV (single monochromator) and MKIII (double monochromator) models, which have been measuring TCO concurrently at the Poprad—Gánovce station since 2014, can also be found there. Moreover, the analysis of the potential impact of the stray-light effect using a statistical method is also included in Appendix C.

# 2.4 AOD calculation

190

The Brewer spectrophotometer at the Poprad-Gánovce station allows for the determination of optical depth in the UV region of the spectrum at wavelengths of 306.3 nm, 310 nm, 313.5 nm, 316.8 nm, and 320 nm. The use of Brewer spectrophotometer measurements to determine AOD has been documented in numerous studies (e.g., Jarosławski et al., 2003; Kazadzis et al., 2007; López-Solano et al., 2018; Nuñez et al., 2023). In this study, the Langley plot method (LPM) was employed to calculate AOD. This traditional method is commonly used to calculate AOD from Brewer spectrophotometer data (Carvalho and Henriques, 2000; Nuñez et al., 2023). Notably, this method of AOD calculation has previously been applied to data from the Poprad-Gánovce site (Pribullová, 2002; Hrabčák, 2018).

This method requires stable atmospheric conditions to determine the extraterrestrial constants (ETCs) with sufficient accuracy. Specifically, low variability in TCO and AOD is essential during the measurement period. It is also necessary to avoid any cloud contamination in DS measurements and to ensure a sufficient range of zenith angles throughout the day, which is crucial for this method. Determining ETCs using the LPM at low-altitude stations in urbanized areas is not recommended unless corrections for the effects of diffuse radiation and the daily ozone cycle are known. This method is most suitable for lower latitudes (particularly in mountainous regions near the tropics) and has certain limitations in the middle and particularly higher latitudes (Marenco, 2007). Poprad-Gánovce is neither a typical low-altitude urbanized station nor a typical high-mountain site, rather it is situated between these two cases (Hrabčák, 2018).

The calculation of AOD is based on the theory described by the Beer-Bouguer-Lambert law. Applying this law to solar UV radiation passing through the atmosphere can be expressed as follows:

$$S_{\lambda} = S_{0,\lambda} e^{-\mu_W \tau_{\lambda}} = S_{0,\lambda} e^{-\mu_{O_3} \tau_{\lambda,O_3} - \mu_r \tau_{\lambda,r} - \mu_a \tau_{\lambda,a}}, \tag{1}$$

where  $S_{\lambda}$  is the flux density of solar radiation flow for the selected wavelength  $\lambda$  expressed by photon count per unit of time at the Earth's surface,  $S_{0,\lambda}$  is the flux density of solar radiation flow for the selected wavelength expressed by photon count per unit of time above Earth's atmosphere (ETC). The total optical depth of atmosphere  $\tau_{\lambda}$  consists of the optical depth for TCO  $\tau_{\lambda,O_3}$ , the optical depth for Rayleigh scattering  $\tau_{\lambda,r}$ , and the AOD is denoted as  $\tau_{\lambda,a}$ , which is the parameter of interest. The contribution of sulfur dioxide was neglected mainly due to its low impact (Arola and Koskela, 2004) and the inaccurate determination at the Poprad-Gánovce station. This site is generally considered rural with low anthropogenic influence, and

SO<sub>2</sub> concentrations are therefore often close to the detection limit. In addition, the O3Brewer software settings for Poprad-195 Gánovce are not well optimised for the reliable retrieval of such low SO<sub>2</sub> values.

Furthermore,  $\mu_{O_3}$  is the air mass factor of the ozone layer determined according to Evans and Komhyr (2008),  $\mu_r$  is the air mass factor for Rayleigh scattering determined according to Kasten and Young (1989),  $\mu_a$  is the air mass factor of aerosols ( $\mu_a 

Figure 1 Daily mean values of TCO (a) and AOD<sub>320</sub> (b) derived from measurements taken by the Brewer ozone spectrophotometer in Poprad-Gánovce from 18 August 1993 to 31 May 2024.

In this section, we present a study on the fundamental behaviour of TCO and AOD over time. The aim is to identify any patterns in these quantities that can be explained by atmospheric circulation or local aerosol emissions. To this end, the daily mean values of TCO and AOD at 320 nm (AOD<sub>320</sub>) are plotted in Figure 1. As observed, both quantities exhibit clear periodic behaviour. As shown in Figure 1a, daily TCO averages vary over a relatively large range. The absolute lowest daily average TCO was recorded on 1 January 1998, with a value of 203 DU. Conversely, the highest daily average was observed on 24 February 1999, with a value of 509 DU. For AOD<sub>320</sub>, all values equal to or greater than 1.5 were excluded during quality control.

The intra-annual variation of TCO and AOD<sub>320</sub> has been analysed. For this purpose, several statistical parameters have been determined, including the median, mean, standard deviation, first and third quartiles, and minimum and maximum values for each month. Figure 2 shows the results summarised in the boxplots. Additionally, these statistics are presented in Table 1 and Table 2. These results are based on monthly means. Clear seasonal patterns are evident in both plots. The annual TCO cycle is more pronounced, with peak mean values occurring in March, followed by a decline to minimum values in October.

The observed annual cycle is driven by the Brewer-Dobson circulation, which plays a key role in the global distribution of TCO (Rosenlof, 1995; Butchart et al., 2006).

Figure 2 Boxplots showing the statistical distribution of the tropopause height (left vertical axis) together with the TCO (right vertical axis) (a) and  $AOD_{320}$  (b) for each month based on data from September 1993 to May 2024. The means are represented by solid points. The horizontal lines inside the boxes indicate the medians. The boxes extend from the 25th percentile (U25) to the 75th percentile (U75). Additionally, the lower and upper whiskers represent the corresponding minimum and maximum values, respectively.




Table 1 Statistical parameters quantifying the inter-annual behaviour of TCO based on data from September 1993 to May 2024. The second column represents the mean, the third the corresponding standard deviation, the fourth the median, followed by U25 and U75, which denote the 25th and 75th percentiles, respectively. N represents the number of days, and M the number of months with available data.

| Month     | TCO (DU) | $\sigma(TCO)(DU)$ | < TCO > (DU) | U25 (DU) | U75 (DU) | N   | М  |
|-----------|----------|-------------------|--------------|----------|----------|-----|----|
| January   | 335      | 17                | 336          | 325      | 344      | 665 | 31 |
| February  | 360      | 20                | 360          | 340      | 370      | 674 | 31 |
| March     | 366      | 16                | 366          | 353      | 379      | 801 | 31 |
| April     | 364      | 16                | 367          | 352      | 374      | 808 | 31 |
| May       | 352      | 12                | 352          | 342      | 359      | 831 | 31 |
| June      | 334      | 8                 | 334          | 326      | 340      | 819 | 30 |
| July      | 322      | 8                 | 321          | 315      | 328      | 873 | 30 |
| August    | 306      | 9                 | 305          | 299      | 313      | 853 | 31 |
| September | 294      | 11                | 292          | 286      | 298      | 784 | 31 |
| October   | 281      | 9                 | 281          | 275      | 286      | 763 | 31 |
| November  | 289      | 11                | 288          | 283      | 293      | 647 | 31 |
| December  | 307      | 15                | 306          | 299      | 313      | 572 | 31 |

In Figure 2a (left), the inter-annual variation of the tropopause height, computed from monthly means, is shown alongside the annual cycle of TCO. The left vertical axis represents tropopause height (m), while the right vertical axis depicts TCO (DU). Both quantities exhibit an approximately opposite behaviour: mean TCO values peak in March, coinciding with the

minimum in tropopause height. The lowest TCO values occur in October, while the maximum tropopause height is observed in August. We can therefore conclude that the second key factor influencing TCO is the tropopause height.

Regarding the mean values of AOD<sub>320</sub>, two peaks are typically observed throughout the year. The first occurs in April, while the second, more pronounced peak, appears in August. The April maximum may be related to the increased occurrence of dust intrusions from the Sahara, which are most frequent from April to June (Hrabčák, 2022). This effect is also found in other European sites in the period March to April (Garcia-Suñer et al., 2024). Additionally, anthropogenic spring biomass burning may also contribute. The highest AOD<sub>320</sub> values during August may be related to characteristic sunny, light wind weather, which favor aerosol accumulation and more effective vertical mixing, increasing the boundary layer thickness. Moreover, due to higher summer irradiation and consequently increased evaporation, columnar water vapor (CWV) levels rise, favoring the growth of hygroscopic particles and thus increasing AOD<sub>320</sub>. Additionally, higher irradiance promotes the formation of secondary aerosols. It is also important to take into account that the influence of local aerosol sources (natural or anthropogenic) is not negligible during the warm half-year, when agricultural activities occur and pollen levels in the air increase.





Table 2 Statistical parameters quantifying the inter-annual behaviour of AOD<sub>320</sub> based on data from September 1993 to May 2024. The second column represents the mean, the third the corresponding standard deviation, the fourth the median, followed by *U*25 and *U*75, which denote the 25th and 75th percentiles, respectively. *N* represents the number of days, and *M* the number of months with available data.

| Month     | AOD <sub>320</sub> | $\sigma(AOD_{320})$ | $< AOD_{320} >$ | <i>U</i> 25 | <i>U</i> 75 | N   | М  |
|-----------|--------------------|---------------------|-----------------|-------------|-------------|-----|----|
| January   | 0.16               | 0.05                | 0.16            | 0.13        | 0.18        | 456 | 31 |
| February  | 0.19               | 0.07                | 0.19            | 0.13        | 0.25        | 474 | 31 |
| March     | 0.26               | 0.09                | 0.25            | 0.21        | 0.29        | 608 | 31 |
| April     | 0.36               | 0.11                | 0.36            | 0.26        | 0.43        | 628 | 31 |
| May       | 0.32               | 0.08                | 0.31            | 0.26        | 0.38        | 669 | 31 |
| June      | 0.33               | 0.08                | 0.30            | 0.28        | 0.39        | 682 | 30 |
| July      | 0.36               | 0.09                | 0.35            | 0.29        | 0.41        | 728 | 30 |
| August    | 0.39               | 0.11                | 0.36            | 0.31        | 0.46        | 733 | 31 |
| September | 0.30               | 0.10                | 0.31            | 0.21        | 0.37        | 642 | 31 |
| October   | 0.21               | 0.07                | 0.20            | 0.16        | 0.23        | 606 | 31 |
| November  | 0.18               | 0.06                | 0.20            | 0.13        | 0.22        | 461 | 31 |
| December  | 0.15               | 0.07                | 0.13            | 0.11        | 0.18        | 387 | 31 |

The lower aerosol loading observed during the cold season can be attributed to various factors. On the one hand, during the cold half-year, a predominant westerly flow from the Atlantic Ocean is observed, bringing clean marine air masses. In addition, the station's higher-altitude location in a windy area prevents the accumulation of anthropogenic aerosols. Furthermore, the terrain surrounding the station is often wet or snow-covered during the cold seasons, which rules out the possibility of local dust sources.

#### 3.2 Analysis of the trends in TCO and AOD






In this section, the time series of TCO and AOD<sub>320</sub> over the years is analysed to determine the temporal evolution of these variables. As a first step, linear trends in the annual and seasonal means of TCO and AOD<sub>320</sub> were determined to identify possible long-term changes. The data points were fitted using the linear regression method, which is used to draw conclusions about their behaviour. The next step is to apply a more advanced analysis: the Mann-Kendall test (Mann, 1945; Kendall, 1948; Gilbert, 1987), which assesses the statistical significance of the trend, and Sen's slope (Sen, 1968), which quantifies the magnitude of the increasing or decreasing trend.

The data for TCO and AOD<sub>320</sub> were analysed by year and also by meteorological seasons. In this way, spring includes data from March to May; summer from June to August; autumn from September to November; and winter from December to February. Figures 3a and 3b show the corresponding linear fits, and Table 3 summarizes the values of the parameters obtained from the fit for TCO and AOD<sub>320</sub>. Due to shorter days, and a lower Sun elevation, fewer days with available measurements are observed during the cold half-year, especially from November to January. Therefore, to appropriately account for the contribution of each month, weighted means were used to calculate annual and seasonal averages. The weights were based on the number of calendar days in each month.

Figure 3 Seasonal and annual trends in TCO (a) and AOD<sub>320</sub> (b) based on data from 1994 to 2023, using linear regressions represented by dashed (seasonal) and solid (annual) lines, respectively. Triangles, squares, hexagons, and diamonds indicate the weighted means for spring, summer, autumn, and winter, respectively. Circles represent the weighted annual means.

Figure 3a shows that the TCO does not follow any clear linear trend, with the exception of summer. On the other hand, focusing on Figure 3b, AOD<sub>320</sub> clearly shows a decreasing trend across all seasons as well as in the annual data. As seen in Table 3, the R<sup>2</sup> values, which are indicators of the goodness of the fit, are quite small for TCO. In the case of AOD<sub>320</sub>, these values are higher, as the corresponding means follow a clear decreasing trend. The trend uncertainty is represented by the standard error of the slope, as provided by the OLS algorithm. In the case of TCO, the uncertainties are larger than the slope

values (except in summer). This, together with the large p-values obtained confirms the statistical insignificance of the seasonal and annual TCO trends. In contrast, the p-values obtained for the seasonal and annual AOD<sub>320</sub> indicate highly significant trends.

Table 3 Parameters (trend, p-value and R<sup>2</sup>) obtained from the linear regression analysis of seasonal and annual TCO and AOD<sub>320</sub>, based on weighted means from 1994 to 2023.

|        | TCO               | )    |        | AOD <sub>320</sub>            |                       |       |  |
|--------|-------------------|------|--------|-------------------------------|-----------------------|-------|--|
| Season | Trend (DU/decade) | p    | $R^2$  | Trend (decade <sup>-1</sup> ) | p                     | $R^2$ |  |
| Spring | 0 ± 2             | 0.83 | 0.002  | $-0.074 \pm 0.009$            | $1.5 \times 10^{-8}$  | 0.69  |  |
| Summer | $-1.6 \pm 1.3$    | 0.24 | 0.05   | $-0.068 \pm 0.011$            | $1.7 \times 10^{-6}$  | 0.56  |  |
| Autumn | $-0.5 \pm 1.7$    | 0.77 | 0.003  | $-0.053 \pm 0.009$            | $9.4 \times 10^{-7}$  | 0.58  |  |
| Winter | 1 ± 3             | 0.71 | 0.005  | $-0.032 \pm 0.008$            | $2 \times 10^{-4}$    | 0.39  |  |
| Annual | $-0.2 \pm 1.4$    | 0.91 | 0.0005 | $-0.057 \pm 0.005$            | $8.6 \times 10^{-12}$ | 0.82  |  |

The decrease in AOD<sub>320</sub> is consistent with the analysis performed by Li et al. (2014), who detected a decreasing AOD trend in European countries. Our results are also consistent with the findings of Filonchyk et al. (2020), who, based on MODIS-Aqua measurements, reported a trend of -0.041 per decade in AOD<sub>550</sub> over Slovakia during the period 2002–2019. This behaviour is primarily attributed to social and industrial changes in the Eastern European region. The key factors include the introduction of governmental policies to reduce anthropogenic pollutant emissions, the limitation of heavy industry, increased gasification, and public education. It is noteworthy that the lowest annual average of AOD<sub>320</sub> in Poprad-Gánovce was measured in 2020, during the first year of the COVID-19 pandemic.

Finally, Table 4 summarizes the results obtained from fitting the time evolution of AOD<sub>320</sub> for each month of the year using the linear regression method. The trend is decreasing for all months. Furthermore, the trends for the months from March to September are steeper than those for the rest of the year. Regarding the p-values, all months indicate significant trends. The results of the Mann-Kendall test and the corresponding values of Sen's slope have also been included in Table 4. It is worth noting that all months exhibit statistically significant AOD<sub>320</sub> decreasing trends at least at the 95 % confidence level. Therefore, it can be concluded that AOD<sub>320</sub> significantly decreases over the years. It is important to note the very good agreement found for each month between Sen's slope values and the slopes obtained from the linear regression analysis.





Table 4 Parameters (trend, p-value and  $R^2$ ) obtained from the linear regression analysis of AOD<sub>320</sub> for each month of the year, based on data from September 1993 to May 2024. In addition, results from the Mann-Kendall test are also included. Specifically, Z represents the test statistic of the Mann-Kendall test, and S denotes Sen's slope. Z values indicating statistically significant trends at the 95 % confidence level (|Z| > 1.96) have been highlighted in bold.

|           | Linear re                     | gression             |       | Mann | Mann-Kendall test         |  |  |
|-----------|-------------------------------|----------------------|-------|------|---------------------------|--|--|
| Month     | Trend (decade <sup>-1</sup> ) | p                    | $R^2$ | Z    | S (decade <sup>-1</sup> ) |  |  |
| January   | $-0.021 \pm 0.009$            | 0.028                | 0.16  | -2.2 | $-0.018 \pm 0.012$        |  |  |
| February  | $-0.043 \pm 0.012$            | $1.7 \times 10^{-3}$ | 0.30  | -2.8 | $-0.044 \pm 0.019$        |  |  |
| March     | $-0.069 \pm 0.015$            | $1.1 \times 10^{-4}$ | 0.42  | -4.2 | $-0.060 \pm 0.019$        |  |  |
| April     | $-0.085 \pm 0.016$            | $1.6 \times 10^{-5}$ | 0.49  | -3.8 | $-0.09 \pm 0.03$          |  |  |
| May       | $-0.067 \pm 0.011$            | $1.5 \times 10^{-6}$ | 0.57  | -4.6 | $-0.072 \pm 0.013$        |  |  |
| June      | $-0.059 \pm 0.013$            | $9.8 \times 10^{-5}$ | 0.42  | -3.2 | $-0.05 \pm 0.02$          |  |  |
| July      | $-0.062 \pm 0.017$            | 9 × 10 <sup>-4</sup> | 0.33  | -3.1 | $-0.06 \pm 0.02$          |  |  |
| August    | $-0.084 \pm 0.019$            | $1.1 \times 10^{-4}$ | 0.42  | -3.4 | $-0.08 \pm 0.03$          |  |  |
| September | $-0.070 \pm 0.018$            | $4.7 \times 10^{-4}$ | 0.36  | -3.6 | $-0.07 \pm 0.02$          |  |  |
| October   | $-0.052 \pm 0.010$            | $1.1 \times 10^{-5}$ | 0.50  | -4.2 | $-0.049 \pm 0.011$        |  |  |
| November  | $-0.039 \pm 0.009$            | $1.5 \times 10^{-4}$ | 0.41  | -3.1 | $-0.036 \pm 0.014$        |  |  |
| December  | $-0.032 \pm 0.013$            | 0.017                | 0.19  | -2.1 | $-0.023 \pm 0.014$        |  |  |

# 3.3 Analysis of the dependence of TCO on tropopause height


Several studies have focused on assessing the extent to which changes in TCO depend on variations in tropopause height (Steinbrecht et al. 1998, Varotsos et al. 2004, Coldewey-Egbers et al. 2022). In all mentioned studies, increasing trends have been found for tropopause height, while trends in TCO exhibit the opposite behaviour. In this section, the methodology used in Steinbrecht et al. (1998) and Varotsos et al. (2004) was applied to the data collected in Poprad-Gánovce. These studies focused on locations in the Northern Hemisphere: Steinbrecht et al. (1998) in Hohenpeissenberg, Germany, and Varotsos et al. (2004) in Athens, Greece, so similar results can be expected for Poprad-Gánovce. However, the main difference between these studies and the present study is that measurements at Poprad-Gánovce cover the period from 1993 to 2024, whereas Steinbrecht et al. (1998) analysed data from 1967 to 1997, and Varotsos et al. (2004) from 1984 to 2002. Therefore, some differences are to be expected, especially in the trends of tropopause height and TCO.

Figure 4 shows the frequency distribution of the measured tropopause height values. The occurrence frequency is calculated by dividing the number of data points in each bin by the total number of data points. Each bin was defined as a half-open interval of height levels; for example, total ozone values contributing to the 8 km bin correspond to tropopause heights between 8.0 km (inclusive) and 9.0 km (exclusive). To account for the seasonality of the tropopause height, data corresponding to May/June/July and November/December/January (Nov/Dec/Jan) are plotted separately. Thus, it can be observed that the distribution for Nov/Dec/Jan is slightly wider than that for May/June/July.

This was also reported by Steinbrecht et al. (1998), who attributed it to more variable weather conditions in winter, which is likely to be the case for Poprad-Gánovce as well. Furthermore, as observed in these studies, the May/June/July distribution is shifted to higher altitudes. Indeed, the corresponding maximum occurs at 12 000 m, while for the Nov/Dec/Jan data, the maximum occurs at 11 000 m. Similar results were found in Athens by Varotsos et al. (2004), although they observed for May/June/July a secondary maximum at 16, 000 m, possibly due to the influence of tropical air.

Figure 4 Occurrence frequency of tropopause heights for May/June/July (red) and November/December/January (purple) periods, based on measurements taken from 18 August 1993 to 31 May 2024.





When analyzing daily mean data, a clear anti-correlation between the two parameters can be observed: local minima in TCO correspond to maxima in tropopause height, and vice versa. This behaviour is consistent with expectations for Eastern Europe (Coldewey-Egbers et al. 2022). To further investigate the relationship between TCO and tropopause height, the methodology used by Steinbrecht et al. (1998) was followed. Figure 5 shows mean TCO values for May/June/July and November/December/January, plotted separately as a function of tropopause height levels. The data used for the plot correspond to daily means between 18 August 1993 and 31 May 2024. The levels were defined in the same manner as in Figure 4; for instance, tropopause height values at 8 km represent the mean of the TCO values measured for tropopause heights in the interval [8, 9) km, and so on. The points were fitted using linear regression analysis to parameterize their relationship.

Table 5 summarises the results of the linear regression analysis performed to quantify the seasonal dependence of TCO on tropopause height. The slope, p-values, and  $R^2$  values from the fit are provided. Based on the  $R^2$  and p-values, a statistically significant inverse relationship between TCO and tropopause height was identified. The slopes are of the same order as those computed using data from Hohenpeissenberg and Athens. In fact, for the May/June/July period, Steinbrecht et al. (1998) reported a decrease in TCO of 16.3 DU/km in Hohenpeissenberg, while Varotsos et al. (2004) found that TCO decreases by  $8.5 \pm 0.7$  DU/km in Athens. In Poprad-Gánovce, the decrease is  $11.5 \pm 0.6$  DU/km. On the other hand, the decrease is slightly steeper in the Nov/Dec/Jan period (11.7  $\pm$  1.0 DU/km). The magnitude of the May/June/July–Nov/Dec/Jan

difference in TCO-tropopause height dependence in Poprad-Gánovce is similar to that in Hohenpeissenberg, where a decrease of 15.7 DU/km was observed in Nov/Dec/Jan. Conversely, this difference is more significant in Athens, where the reported rate is –11.2 ± 0.5 DU/km for Nov/Dec/Jan. In any case, the slope is always negative, meaning that TCO decreases with the increasing height of the tropopause.

Figure 5 Linear fit of TCO means obtained for different ranges of tropopause height during the May, June, and July (purple) and November, December, and January (red) periods. The data set considered for the plot corresponds to days between 18 August 1993 and 31 May 2024, when daily means for both tropopause height and TCO are available.

Table 5 Parameters (slope, p-value and R<sup>2</sup>) resulting from the linear regression analysis of TCO means computed for different ranges of tropopause height, based on data from 18 August 1993 to 31 May 2024.

| Months        | Slope (DU/km)   | p                    | $R^2$ |
|---------------|-----------------|----------------------|-------|
| May/June/July | $-11.5 \pm 0.6$ | $8.2 \times 10^{-9}$ | 0.98  |
| Nov/Dec/Jan   | $-11.7 \pm 1.0$ | $3.0 \times 10^{-6}$ | 0.94  |

The temporal evolution of deseasonalized monthly running means of the TCO (ΔTCO) and tropopause height (Δh) time series over nearly 31 years of measurements is presented in Figure 6a and 6b, respectively. Deseasonalization was achieved by computing the difference between each monthly mean and the long-term mean for the corresponding calendar month over the period 1993–2024. The differences were next smoothed by applying a 13-month running mean. Hence, note that although data from September 1993 to May 2024 were available, running means can only be computed from March 1994 to November 2023.

Figure 6 Representation of the time evolution of deseasonalized monthly running means of TCO (a) and tropopause height (b) from March 1994 to November 2023 (circles). Dashed lines represent linear fits to the data.

Next, linear regression was applied to the differences. The linear trend for  $\Delta TCO$  is  $0.0 \pm 0.4$  DU per decade, and for  $\Delta h$ , it is  $105 \pm 7$  m per decade. Our results (particularly for  $\Delta TCO$ ) differ significantly from those reported by Steinbrecht et al. (1998) [ $\Delta TCO = -10$  DU/decade;  $\Delta h = 150$  m/decade] and Varotsos et al. (2004) [ $\Delta TCO = -7.5 \pm 1.0$  DU/decade;  $\Delta h = 167 \pm 30$  m/decade]. However, whereas Steinbrecht et al. (1998) analysed data from 1967–1997 and Varotsos et al. (2004) from 1984–2002, measurements in Poprad-Gánovce cover a more recent period (1994–2023). It is well known that various measures and regulations regarding the production of stratospheric ozone-depleting substances have been implemented to restore the ozone layer, most notably the Montreal Protocol in 1987 and its subsequent amendments. The effects of these actions have become evident in the halting of the TCO decline and the gradual recovery of TCO in recent years.

Following the investigations of Steinbrecht et al. (1998) and Varotsos et al. (2004), the linear relationship between TCO and tropopause height indicates a TCO decline of approximately –12 DU per 1 km increase in tropopause altitude. However, this relationship must be interpreted with caution when applied to different time periods. Indeed, several factors influence the relationship between TCO and tropopause height, including changes in the chemical composition of the atmosphere, which affect TCO, as well as stratospheric cooling and changes in the Brewer-Dobson circulation associated with climate change. The increase in tropopause height, primarily related to rising temperatures in the troposphere due to increased concentrations of greenhouse gases (Meng et al. 2021), may contribute to ozone depletion by shifting the ozone layer to higher altitudes (Match et al. 2022). This subsequently triggers photochemical processes that are enhanced at higher altitudes (Brasseur and Solomon, 1984), which in turn consume ozone and lead to a decrease in TCO (Steinbrecht et al., 1998). When applying the above-mentioned relationship between TCO and tropopause height, an increase in tropopause height of 105 m per decade would correspond to a decrease in TCO of approximately –1.2 DU per decade. However, in reality, the TCO at Poprad-Gánovce does not follow this trend; instead, it shows no clear trend at a rate of 0.0 ± 0.4 DU per decade (Figure 6a).

Table 6 Parameters (trend, p-value and R<sup>2</sup>) obtained from the linear regression analysis of tropopause height and TCO for each month of the year, based on data from January 1994 to December 2023. Additionally, results from the Mann-Kendall test are included. Specifically, Z represents the Mann-Kendall test statistic, and S denotes Sen's slope. Z values indicating statistically significant trends at a confidence level of at least 95 % (|Z| > 1.96) are highlighted in bold.

|           |                     | Trop      | popause l | height            |                 |                      |                   | тсо    |      |                   |  |  |  |
|-----------|---------------------|-----------|-----------|-------------------|-----------------|----------------------|-------------------|--------|------|-------------------|--|--|--|
|           | Linear              | regressio | on        | Mann-Kendall test |                 | Linear               | Linear regression |        |      | Mann-Kendall test |  |  |  |
| Month     | Trend<br>(m/decade) | p         | $R^2$     | Z                 | S<br>(m/decade) | Trend<br>(DU/decade) | p                 | $R^2$  | Z    | S<br>(DU/decade)  |  |  |  |
| January   | $-60 \pm 90$        | 0.50      | 0.016     | -1.0              | $-110 \pm 140$  | 5 ± 3                | 0.19              | 0.06   | 1.2  | 4 ± 5             |  |  |  |
| February  | $70 \pm 120$        | 0.59      | 0.011     | 0.8               | $100 \pm 140$   | $-2 \pm 5$           | 0.73              | 0.004  | -0.5 | $-4 \pm 7$        |  |  |  |
| March     | $90 \pm 80$         | 0.28      | 0.04      | 1.0               | $90 \pm 130$    | 2 ± 3                | 0.47              | 0.018  | 0.6  | 2 ± 5             |  |  |  |
| April     | $80 \pm 90$         | 0.37      | 0.03      | 0.8               | $100 \pm 120$   | $-3 \pm 3$           | 0.43              | 0.02   | -0.9 | $-3 \pm 4$        |  |  |  |
| May       | $-30 \pm 80$        | 0.70      | 0.005     | -0.4              | $-20 \pm 100$   | 1 ± 2                | 0.58              | 0.011  | 1.2  | 3 ± 3             |  |  |  |
| June      | $130 \pm 60$        | 0.04      | 0.14      | 2.2               | $150 \pm 80$    | $-0.4 \pm 1.7$       | 0.80              | 0.002  | -0.5 | $-1 \pm 3$        |  |  |  |
| July      | $120 \pm 80$        | 0.13      | 0.08      | 1.6               | 200 ± 100       | $-0.7 \pm 1.6$       | 0.65              | 0.008  | -0.5 | $-1 \pm 2$        |  |  |  |
| August    | $200 \pm 70$        | 0.008     | 0.23      | 2.9               | $200 \pm 100$   | $-3.6 \pm 1.8$       | 0.06              | 0.12   | -1.8 | $-4 \pm 2$        |  |  |  |
| September | $200 \pm 100$       | 0.04      | 0.14      | 2.1               | $250 \pm 150$   | $-3 \pm 2$           | 0.23              | 0.05   | -1.4 | $-3 \pm 3$        |  |  |  |
| October   | $100 \pm 100$       | 0.19      | 0.06      | 1.5               | $200 \pm 130$   | $-0.8 \pm 1.9$       | 0.69              | 0.006  | 0.1  | 0 ± 3             |  |  |  |
| November  | $150 \pm 90$        | 0.10      | 0.09      | 1.9               | $200 \pm 100$   | 2 ± 2                | 0.34              | 0.03   | 1.0  | 2 ± 3             |  |  |  |
| December  | $90 \pm 100$        | 0.41      | 0.02      | 0.7               | $70 \pm 130$    | $0\pm3$              | 0.99              | 5.10-6 | -0.2 | $0 \pm 4$         |  |  |  |

Although no clear impact of the increasing trend in tropopause height on TCO is evident in the deseasonalized monthly running means, distinct variations are observed in certain months. The temporal evolution of the monthly mean of tropopause height and TCO has been analysed using linear regression. The corresponding results are summarized in Table 6, in a manner similar to those presented in Table 4 for AOD<sub>320</sub>. Although not all months exhibit the expected behaviour (e.g., January and May for tropopause height), it is important to notice that for the best fits the expected trend is found. The corresponding errors were also determined for the linear trends. For tropopause height, the errors exceed the slope values in January, February, April, May, and December. The corresponding p-values and R<sup>2</sup> from the linear regression analysis indicate that the trends are not statistically significant for these months. For the remaining months, high p-values are also obtained in July, October, and November, whereas significant dependencies are found in June, August, and September. With regard to TCO, and focusing on the p-values from the linear regression analysis, only the trend in August can be considered significant at the 90 % confidence level. For the remainder of the year, the p-values are very high, indicating statistically insignificant trends.

The Mann-Kendall test was also applied to analyse trends in the monthly data. It is worth noting that the slopes determined from the linear fits are very similar to the Sen's slopes, confirming the consistency between the two methods. Significant increasing trends in tropopause height at a confidence level of at least 95 % have been found in June, August, and September in agreement with the linear regression results. No statistically significant relationship was found for TCO at the 95 %


confidence level. However, weak but not statistically significant declining trends in TCO were identified in August. It is precisely in the case of this month that it can be argued that the observed decreasing trend in TCO is likely caused by a significant increase in tropopause height. An indication of a downward trend in TCO in April may be primarily related to anthropogenic stratospheric ozone depletion. Episodes of unusually low TCO values were observed in April 2011 and 2020, following exceptionally cold and prolonged stratospheric winters, characterized by a strong and persistent Arctic polar vortex (Manney et al., 2020). On the other hand, the indication of an upward trend in TCO in January, in addition to the weakening influence of halogen gases and a slight decrease in tropopause height, suggests a potential acceleration of the Brewer-Dobson circulation, which is most active during the local winter.

#### 3.4 Evaluation of the hypothetical TCO






The obtained hypothetical TCO time series was deseasonalised and smoothed. The description of the procedure used to separate the contribution of tropopause height is provided in Appendix D. The results are presented in Figure 7a. The dashed line in the plot represents the linear fit to the data, with a slightly positive slope of  $1.3 \pm 0.3$  DU per decade. For completeness, Figure 7b illustrates the corresponding temporal evolution of TCO attributed to changes in tropopause height. As expected, the slope is negative,  $-1.3 \pm 0.1$  DU per decade. It should be noted that its absolute value is similar to that of the slope obtained for the hypothetical TCO, to which factors such as natural atmospheric cycles, additional climate change influences, and the presence of ODS contribute.

The temporal evolution of the hypothetical  $(TCO_{hypo})$  and tropopause-related  $(TCO_{tropo})$  TCO series has also been examined by analysing their long-term trends for each month, analogous to the results shown in Table 6. Both linear regression and the Mann-Kendall test combined with Sen's slope estimator have been applied. The results are presented in Table 7. When comparing the monthly trends of the TCO (Table 6) with those of the hypothetical TCO (Table 7), a weakly increasing trend is evident in the former only in January, whereas in the latter it also appears in March and November. It is noteworthy that not all of the decreasing trend in TCO in August can be explained solely by the tropopause increase, as the  $TCO_{hypo}$  trend for this month remains negative. Another reason for the decrease in TCO in August could be related to changes in large-scale circulation patterns.

Figure 7 Time series of deseasonalised monthly running means of (a) hypothetical TCO and (b) TCO related to tropopause height from March 1994 to November 2023 (circles). Dashed lines indicate linear fits to the data.

Table 7 Parameters (trend, p-value and R²) obtained from the linear regression analysis of hypothetical and tropopause TCO for each month of the year, based on data from January 1994 to December 2023. Additionally, results from the Mann-Kendall test are included. Specifically, **Z** represents the Mann-Kendall test statistic, and **S** denotes Sen's slope. **Z** values indicating statistically significant trends at a confidence level of at least 95 % (|**Z**| > 1.96) are highlighted in bold.

|           |                      |                                     | TCO <sub>hypo</sub>  |        |                              |                      | ı     | TCO <sub>trope</sub> | )     | S<br>(DU/decade) |  |  |
|-----------|----------------------|-------------------------------------|----------------------|--------|------------------------------|----------------------|-------|----------------------|-------|------------------|--|--|
|           | Linear               | Linear regression Mann-Kendall test |                      | Linear | regression Mann-Kendall test |                      |       | Kendall test         |       |                  |  |  |
| Month     | Trend<br>(DU/decade) | p                                   | $R^2$                | Z      | S<br>(DU/decade)             | Trend<br>(DU/decade) | p     | $R^2$                | Z     |                  |  |  |
| January   | $4\pm3$              | 0.19                                | 0.06                 | 1.2    | $4 \pm 4$                    | $0.6 \pm 1.1$        | 0.63  | 0.009                | 0.4   | $0.5 \pm 1.8$    |  |  |
| February  | $0 \pm 4$            | 0.92                                | 0.0004               | -0.2   | $-1 \pm 5$                   | $-1 \pm 2$           | 0.51  | 0.016                | -0.9  | $-2 \pm 3$       |  |  |
| March     | $4\pm3$              | 0.14                                | 0.08                 | 1.6    | 5 ± 3                        | $-2.0 \pm 1.5$       | 0.19  | 0.06                 | -1.2  | $-2 \pm 2$       |  |  |
| April     | $-1 \pm 3$           | 0.66                                | 0.007                | -0.6   | $-1 \pm 4$                   | $-1.4 \pm 1.5$       | 0.34  | 0.03                 | -0.9  | $-2 \pm 2$       |  |  |
| May       | 1 ± 2                | 0.74                                | 0.004                | 0.6    | 2 ± 3                        | $0.7 \pm 0.9$        | 0.47  | 0.019                | 0.8   | $0.9 \pm 1.3$    |  |  |
| June      | $1.2 \pm 1.7$        | 0.48                                | 0.018                | 0.5    | 1 ± 3                        | $-1.7 \pm 0.7$       | 0.025 | 0.17                 | -1.96 | $-1.5 \pm 1.1$   |  |  |
| July      | $0.1\pm1.5$          | 0.97                                | 4 × 10 <sup>-5</sup> | -0.4   | $-0.4 \pm 1.7$               | $-1.1 \pm 0.9$       | 0.26  | 0.05                 | -1.14 | $-1.2 \pm 1.3$   |  |  |
| August    | $-2.7 \pm 1.7$       | 0.12                                | 0.08                 | -1.5   | $-3 \pm 2$                   | $-1.3 \pm 0.5$       | 0.011 | 0.21                 | -2.5  | $-1.4 \pm 0.6$   |  |  |
| September | $-1.2 \pm 1.8$       | 0.52                                | 0.015                | -1.0   | $-1 \pm 2$                   | $-1.6 \pm 0.7$       | 0.023 | 0.17                 | -2.7  | $-2.0 \pm 0.9$   |  |  |
| October   | $0.5\pm1.5$          | 0.66                                | 0.004                | 0.6    | 1 ± 2                        | $-1.3 \pm 0.7$       | 0.059 | 0.12                 | -1.8  | $-1.4 \pm 1.0$   |  |  |
| November  | $4.5 \pm 1.7$        | 0.015                               | 0.20                 | 1.93   | 4 ± 2                        | $-2.3 \pm 1.2$       | 0.063 | 0.12                 | -2.1  | $-2.3 \pm 1.6$   |  |  |
| December  | 2 ± 2                | 0.50                                | 0.017                | 0.7    | 2 ± 3                        | $-1.7 \pm 1.6$       | 0.28  | 0.04                 | -1.0  | $-1.6 \pm 1.8$   |  |  |

When comparing the results of both approaches, agreement between the linear regression slopes and Sen's slopes is evident.

Furthermore, it is important to mention that no statistically significant trend has been found except in November, where the corresponding p-value in the linear regression analysis is 0.015. For the other months, p-values are quite high. Regarding

 $TCO_{tropo}$ , the trends are negative or close to 0, as expected, showing the strong correlation between the decrease in TCO and the increase in tropopause height. In this case, both statistical approaches revealed the strongest statistically significant trends in August and September. Finally, Figure 8 clearly illustrates the findings discussed above. The hypothetical TCO shows a positive evolution over time, while the  $TCO_{tropo}$  trend is negative. When combined, the overall trend is slightly positive, but statistically insignificant. Therefore, it can be concluded that atmospheric TCO is primarily governed by two components: a decrease linked to the rising tropopause height ( $TCO_{tropo}$ ) and an increase related to  $TCO_{hypo}$ . The second factor can be attributed, on the one hand, to the implementation of policies aimed at reducing ODS emissions. On the other hand, the acceleration of the Brewer-Dobson circulation due to climate change probably contributes to the increase in TCO by enhancing the transport of ozone to mid-latitude sites such as Poprad-Gánovce.

Figure 8 Representation of the time evolution of deseasonalised monthly running means of hypothetical (fuchsia), tropopause (blue) and observed (green) TCO from March 1994 to November 2023. Dashed lines represent linear fits to the data.

# 530 3.5 LOTUS regression analysis




The results of the LOTUS regression analysis are presented in Table 8. K will not be taken into account, given its non-physical meaning. Three predictors show a highly statistical significant trend: ENSO, QBOB and HEIGHT. Among the predictors, HEIGHT exerts the greatest influence on TCO. As expected, the trend is negative: the decrease in TCO may be associated with an increase in tropopause height. The effect of ENSO on TCO is positive, i.e. contributing to an increase in TCO. This has also been confirmed by other studies (e.g. Zhang et al. 2015, Li et al. 2024). The effect of the B component of the QBO, which is zonally asymmetric according to Wang et al. (2022), translates into a decrease in TCO. Conversely, a smaller (in absolute value) but positive slope for linear\_post could be interpreted as the existence of an increasing trend in TCO since 1997, which would be consistent with the reduction in ODS emissions into the atmosphere.

Table 8 Results of the LOTUS regression analysis applied to the set of monthly TCO means from measurements taken at the Poprad-Gánovce station covering from September 1993 to February 2024. The analysis yielded a R<sup>2</sup> = 0.88. The slope value is an indicator of the weight of each parameter in the model, while the sign determines its effect on the TCO (positive causes an increase and negative a decrease). CI represents the confidence interval with a confidence level of 95 %. Finally, the p-values indicate the significance of the trend. Those with p < 0.01 (highlighted in bold in the table) show very significant trends.

| Predictor   | Slope (unit <sup>-1</sup> ) | CI             | p-value               |
|-------------|-----------------------------|----------------|-----------------------|
| ENSO        | $1.8 \pm 0.7$               |                | 0.007                 |
| SOLAR       | $0.7 \pm 0.7$               | [-0.7, 2.0]    | 0.341                 |
| QBOA        | $-0.8 \pm 0.7$              | [-2.1, 0.6]    | 0.264                 |
| QBOB        | $-3.1 \pm 0.7$              | [-4.4, -1.7]   | $1.3 \cdot 10^{-5}$   |
| AOD         | $0.4 \pm 1.4$               | [-2.2, 3.1]    | 0.758                 |
| Linear_pre  | $10 \pm 20$                 | [-30, 60]      | 0.515                 |
| Linear_post | $2.0 \pm 1.2$               | [-0.4, 4.3]    | 0.099                 |
| HEIGHT      | $-13.5 \pm 0.8$             | [-15.2, -11.9] | 3.5·10 <sup>-43</sup> |

#### **4 Conclusions**




This study summarizes the results obtained from the climatological analysis of over 30 years (18 August 1993 – 31 May 2024) of TCO and AOD data collected by a Brewer ozone spectrophotometer installed at the Poprad-Gánovce station in Slovakia. In particular, the intra-annual behaviour and the evolution of the annual, seasonal, and monthly means over the years have been analysed. The analysis found that both parameters exhibit seasonal patterns. On the one hand, AOD<sub>320</sub> exhibits two distinct peaks during the warm half-year. The peak in April may be related to Saharan dust intrusions and spring biomass burning, while the one in August could be associated with summer conditions that favour particle accumulation, growth, and the formation of secondary aerosols. Higher TCO levels are observed from February to April, likely associated with the Brewer-Dobson circulation and the height of the tropopause.

Regarding temporal trends, the obtained results are in good agreement with the broader context of recent atmospheric developments in Central and Eastern Europe. No clear trend has been identified for the annual averages of TCO, while AOD<sub>320</sub> has shown a distinct decreasing trend over the years, with a value of  $-0.057 \pm 0.005$  per decade. This is consistent with other studies in Europe (Li et al., 2014; Filonchyk et al., 2020; Garcia-Suñer et al., 2024). It is noteworthy that the decreasing trend in AOD<sub>320</sub> is statistically significant in all months of the year. The explanation lies in the social and industrial changes in the region that have caused a significant decrease in anthropogenic air pollution. It is important to note that the lowest annual average of AOD<sub>320</sub> was measured in 2020, during the first year of the COVID-19 pandemic.

The relationship between TCO and tropopause height has also been examined. In contrast to TCO, which in fact shows no clear linear trend ( $0.0 \pm 0.4$  DU per decade), tropopause height exhibits a pronounced increase of  $105 \pm 7$  m per decade. Two auxiliary TCO trends were calculated. The first represents a hypothetical TCO in which the influence of long-term changes in tropopause height was removed ( $1.3 \pm 0.3$  DU per decade), while the second reflects the temporal evolution of TCO

attributed to variations in tropopause height ( $-1.3 \pm 0.1$  DU per decade). Long-term changes analysed for individual months reveal distinct features. The statistically most significant linear increasing trend in tropopause height has been found in August, reaching  $200 \pm 70$  m per decade. The most pronounced negative linear trend in TCO was also observed in August, amounting to  $-3.6 \pm 1.8$  DU per decade. It was found that not all of the decreasing trend in TCO in August can be explained solely by the increase in tropopause height, but only slightly more than one third of it. Furthermore, a statistically significant positive linear trend was found in November for hypothetical TCO, after the effect of tropopause height increase was removed. A more pronounced influence of the tropopause height increase on TCO was also detected in June and September. In summary, atmospheric TCO appears to be controlled by three main factors: a decrease linked to the rising tropopause height and an increase associated with recovery from ODS reductions and a likely strengthening of the Brewer-Dobson circulation.

Finally, the LOTUS regression analysis was presented. The notable influence of tropopause height on TCO, in terms of an inverse proportionality, was confirmed. A highly statistically significant role of natural atmospheric cycles in determining the temporal evolution of TCO was demonstrated, with a positive influence of ENSO and a negative influence of QBOB. In addition, a positive but statistically less significant slope  $(2.0 \pm 1.2 \text{ DU})$  for TCO since 1997 was found, consistent with the reduction in atmospheric ODS emissions. It can be stated that while the negative impact of halogen gases is gradually weakening, the influence of climate change on TCO is increasing. Therefore, continuing regular measurements of TCO is necessary.

#### Appendix A: Description of selected inputs used in the AOD calculation

#### Multistep raw data post-processing:







The value of  $S_{\lambda}$  from Eq. (1) was obtained by adjustment of raw data (raw counts). The raw data cover the period from 18 August 1993 to 31 May 2024. It is essential to maintain the sequence of the following steps during their adjustment. In the first step, raw counts stored in a B-file were converted into count rates. In the second step, dead time compensation was applied. Both formulas are provided in the Brewer MKIV Spectrophotometer Operator's Manual (SCI-TEC Instruments Inc., 1999). After the dead time compensation, a correction for the stray-light effect was applied in the same manner as in Garane et al. (2006). The methodology of the given correction in the case of Brewer at the Poprad-Gánovce station is described in more detail in Hrabčák (2018).

In the fourth step, a correction for the temperature dependence was applied, including a spectral transmittance correction for the neutral density (ND) filter used (SCI-TEC Instruments Inc., 1999). These filters are automatically selected by the Brewer spectrophotometer based on the current solar radiation flux. There are five ND filters and five wavelengths, resulting in a total of 25 required attenuation values. The attenuation values for the filters are determined during the instrument's calibration. The attenuation values are derived from the filterwheel #2 test and subsequently recorded in the FW2TEST file.

In the fifth step, the impact of polarization was compensated. The sixth and final step involved the correction for the impact of diffuse radiation on DS measurements following the recommendations of Arola and Koskela (2004). The methodology of this correction is described in more detail in Hrabčák (2018).

Applying the above criteria yields five initial values of  $S_{\lambda}$  from a single DS measurement, from which the five AODs is subsequently calculated. The final AOD for the given DS measurement is calculated as the arithmetic average of the five values. All final values of AOD underwent a quality control procedure, including cloud screening, according to Hrabčák (2018).

# TCO optical depth calculation:



The TCO optical depth  $\tau_{\lambda,O_3}$  was calculated for each accepted value of the TCO. The calculation is represented by the following equation:

$$\tau_{\lambda,O_3} = \Omega_{O_3} \alpha(\lambda,T) = \Omega_{O_3} \sigma(\lambda,T) n, \qquad (A1)$$

where  $\Omega_{0_3}$  is the TCO in DU, and the next element  $\alpha(\lambda, T)$ , is the absorption coefficient for ozone, which can be further 610 expressed as the product of  $\sigma(\lambda, T)$ , the effective absorption cross section of the ozone molecule (typically quantified for 1cm<sup>2</sup>), and n, the molecule count given by 1 DU and 1cm<sup>2</sup>. For ozone, this is a constant with the value  $n = 2.687 \times 10^{16} \, \text{DU}^{-1}$ cm<sup>-2</sup> (Schwartz and Warneck, 1995). Carlund et al. (2017) recommend utilizing the same effective absorption cross sections of the ozone molecule to calculate both the TCO and its optical depth, which are required to determine the AOD. The O3Brewer software utilizes the effective absorption cross sections of the ozone molecule, which are based on the 615 measurements by Bass and Paur (1985). This scale is used to calculate the TCO within the Brewer spectrophotometer of network, following the recommendations International Ozone the Commission (http://www.esrl.noaa.gov/gmd/ozwv/dobson/papers/coeffs.html; last accessed: February 2025).

More recent and accurate values of effective absorption cross sections are now available based on measurements from the Molecular Spectroscopy Laboratory at the Institute of Environmental Physics (IUP), University of Bremen (https://www.iup.uni-bremen.de/gruppen/molspec/databases/; last accessed: Febraury 2025; Gorshelev et al., 2014; Serdyuchenko et al., 2014). These values were used to calculate the TCO optical depth. To maintain consistency between the calculation of TCO optical depth and TCO, the TCO values (used exclusively for TCO optical depth calculations) were also calculated using the more recent set from the IUP. This calculation was performed following the recommendations of Redondas et al. (2014). The dependence of the effective absorption cross section on temperature is significant. An effective temperature is typically used for the given gas. In the case of ozone measurements using the Brewer spectrophotometer, a standard effective temperature of –45 °C (228.15 K) is defined (Redondas et al., 2014).



# Rayleigh scattering optical depth calculation:

The Rayleigh scattering optical depth  $\tau_{\lambda,r}$  was calculated by the following equation:

$$\tau_{\lambda,r} = \beta(\lambda) \frac{P}{P_{std}},\tag{A2}$$

where element  $\beta(\lambda)$  is the normalized optical depth for Rayleigh scattering under standard atmospheric pressure and for a vertical column, P is the atmospheric pressure at the observation site (an interpolation based on 3 daily measurements in local climatic terms was used), and  $P_{std}$  is the standard atmospheric pressure (101 325 Pa).

The normalized optical depth for Rayleigh scattering  $\beta(\lambda)$  was calculated according to Bodhaine et al. (1999), specifically for the purpose of AOD calculation. The use of coefficients from Bodhaine et al. (1999) is consistent with the recommendations stated in the National Oceanic and Atmospheric Administration (NOAA) online document (https://www.esrl.noaa.gov/gmd/grad/neubrew/docs/RayleighInBrewer.pdf; last access: February 2025), which specifies that the standard coefficients used in the Brewer operating software must not be used to calculate the AOD. It is stated in Carlund et al. (2017) that the TCO calculated using the standard coefficients is higher compared to the use of coefficients according to Bodhaine et al. (1999). In view of the above and for the sake of consistency, the TCO values (only for the purpose of AOD calculation) were also determined using the coefficients from Bodhaine et al. (1999), instead of the standard coefficients.

#### ETCs calculation






The ETC  $S_{0,\lambda}$  for each wavelength was determined using the LPM. The LPM employs multiple measurements of direct solar radiation at different solar zenith angles. The key is to linearize Eq. (1) by applying the natural logarithm:

$$\ln(S_{\lambda}) = \ln(S_{0,\lambda}) - \mu_w \tau_{\lambda} . \tag{A3}$$

For each measurement of direct solar radiation, there is one corresponding equation, where  $\mu_w$  and  $\ln(S_\lambda)$  are the known variables, while  $\ln(S_{0,\lambda})$  and  $\tau_\lambda$  are the unknowns. It is essential to linearly interpolate the obtained dependence of  $\ln(S_\lambda)$  on  $\mu_w$  using the least squares method.  $\ln(S_{0,\lambda})$  is obtained when  $\mu_w$  equals 0. The ETC for the given wavelength is valid for the entire intercalibration period, which is 2 years, consistent with the standard intercalibration period for ozone measurements. Neglected changes in the sensitivity of the instrument over shorter time periods represent a disadvantage of this method. It is assumed that any significant service modifications to the Brewer spectrophotometer during calibration may affect the calculation of both TOC and AOD. For this reason, a period not exceeding 2 years was used.

The methodology for calculating the ETCs is based on Hrabčák (2018); however, some of the points outlined below have been modified (condition 2: the AOD threshold at 320 nm was reduced from 0.5 to 0.4, condition 6: the standard deviation of the daily TCO was relaxed from < 2.5 DU to < 3 DU, condition 7: the standard deviation of the daily AOD was tightened from < 0.07 to < 0.04, and condition 9: the required determination coefficient of the linear interpolation was increased from

- > 0.98 to > 0.99). The ETCs for individual wavelengths can be determined solely from DS measurements that meet the following three conditions:
- 1. The air mass factor for the atmosphere as a whole is less than 3.
- 2. The AOD calculated as an average of five values within a single DS measurement for the wavelength of 320 nm is less than 0.4.
  - 3. The difference between the maximum and minimum value of AOD within a single DS measurement is less than 0.03. Additionally, the ETCs were determined only for days that meet conditions 4 to 9:
  - 4. The number of direct solar radiation measurements is at least 50 (i.e. 10 DS measurements).
- 5. The difference between the maximum and minimum air mass factor for the entire atmosphere is greater than 1.
  - 6. The standard deviation from the measured values of the TCO on the given day is less than 3 DU.
  - 7. The standard deviation from the measured AOD values on the given day is less than 0.04.
  - 8. The following selection criteria were applied:

$$\left|\ln\left(S_{\lambda}\right)_{i} - \ln\left(S_{\lambda}\right)_{j}\right| 

Figure A1 Time series of ETC values for the 320 nm wavelength during 16 intercalibration periods, from 18 August 1993 to 31 May 2024.

#### Appendix B: Intercomparison of TCO datasets





In this section, the presented TCO dataset is compared with existing observations archived in the WOUDC and the EUBREWNET. It is important to note several remarks at the outset. The EUBREWNET Level 1.5 dataset dates back to 2013, which is considerably later than the dataset used in this study and those archived in the WOUDC. The TCO data for the Poprad-Gánovce station in WOUDC are produced using the mentioned O3Brewer software. This is the same software that was used to calculate the TCO dataset in this study. Older data in WOUDC were calculated using previous versions of the software, and this may lead to small discrepancies, as the software has been continuously developed over the years. Further, it is important to note that the data submitted to WOUDC once per month for the preceding month also include daily averages derived solely from the calculation of TCO using measurements of diffuse solar radiation (so-called ZS measurements). This applies to days when it was not possible to retrieve TCO from DS measurements. ZS measurements are subject to considerable uncertainty. The TCO dataset in this study is derived exclusively from DS measurements. A comparison of these two different data sets may therefore lead to discrepancies, particularly during months with a higher proportion of ZS measurements (November–February).

Hence, two basic comparisons were performed, both based on monthly means. The first represents a straightforward approach, involving the visual comparison of the temporal evolution of monthly TCO means obtained for each dataset (Figure B1). At first glance, it can be said that there is a general agreement between the datasets. However, some discrepancies can be identified at the extremes (maxima and minima) between the calculated monthly averages and those

from WOUDC. Something similar is found when comparing the EUBREWNET TCO means with those of the other two datasets, as EUBREWNET values tend to be higher at the maxima and lower in the minima. To quantitatively assess the discrepancies, several statistical parameters were calculated: the root mean square error (RMSE), the standard deviation of the differences (STD), the mean bias deviation (MBD), and the coefficient of determination (R<sup>2</sup>). These are summarised in Table B1.

The discrepancies between WOUDC and the calculated TCO are the smallest, with an RMSE and STD of the differences of 5.00 DU and 3.42 DU, respectively. In the comparison between EUBREWNET and the calculated TCO, these values are slightly higher (5.13 DU and 3.66 DU, respectively), but much lower than the values corresponding to EUBREWNET/WOUDC, 7.26 DU and 4.67 DU, respectively. However, it should be noted that, in the case of the latter two, the number of points compared is much smaller, as the EUBREWNET dataset begins in 2013. These statistical parameters measure the discrepancies between the compared values, showing less dispersion in the case of the comparisons involving the dataset used in this analysis. The same trend is observed with the MBD, which is −1.90 DU for WOUDC/Calculated, 2.45 DU for EUBREWNET/Calculated and 4.48 DU for EUBREWNET/WOUDC. The negative sign for WOUDC/Calculated indicates that mean TCO for WOUDC are slightly higher than those calculated. Following the same reasoning, the calculated TCO are slightly higher than those of EUBREWNET, and the WOUDC values are consistently higher than those of EUBREWNET. All these parameters are much smaller than the measured TCO (~200–500) confirming the compatibility among the values of the different datasets. Finally, the R²≈1 values in each case indicate the goodness of the fits, supporting the claim regarding the agreement between datasets.

Figure B1. Evolution of monthly mean TCO values derived from measurements by the Brewer ozone spectrophotometer at Poprad-Gánovce. Blue triangles represent values processed by the WOUDC, purple squares indicate products of the EUBREWNET, and golden circles denote the TCO values determined in this study. The WOUDC product (as well as the dataset used in this study) covers the period from August 1993 to May 2024, whereas EUBREWNET Level 1.5 TCO values are available from August 2013 onwards.

Table B1 Statistical parameters (RMSE, STD, MBD and  $R^2$ ) for the quantitative study comparing the TCO of the WOUDC and EUBREWNET networks, as well as the TCO values used in this study.

|                      | RMSE (DU) | STD (DU) | MBD (DU) | $R^2$ |
|----------------------|-----------|----------|----------|-------|
| WOUDC/Calculated     | 5.00      | 3.42     | -1.90    | 0.98  |
| EUBREWNET/Calculated | 5.13      | 3.66     | 2.45     | 0.98  |
| EUBREWNET/WOUDC      | 7.26      | 4.67     | 4.48     | 0.95  |

#### **Appendix C: Intercomparison of Brewer spectrophotometers**

First, a comparison is presented between two Brewer spectrophotometers, the MKIV (single monochromator) and MKIII 745 (double monochromator) models, which have been measuring TCO concurrently at the Poprad-Gánovce station since 2014. Pairs of DS measurements were compared, with a maximum time difference of 10 minutes between them. It was decided to perform the comparison starting from the first calibration of the MKIII model (No. 225) by IOS, which took place in 2015. Table C1 presents the results of individual statistical parameters, which are listed separately for each intercalibration period. 750 We also note that the MKIII model experienced significant problems with the upper of its two micrometers during the first years of operation (until 2018), which may have caused certain deviations in the TCO measurements. This fact is likely reflected in the higher RMSE and standard deviation observed during the first two periods. Because of the technical issue mentioned above, the comparison is of limited relevance (related to the investigation of the stray-light effect) for the first two periods. Looking further at the MBD values for the last three periods, it can be seen that 755 they range from -2 to 2 DU, which is essentially below 1 % relative to the typical TCO values. The Brewer MKIV Spectrophotometer Operator's Manual (SCI-TEC Instruments Inc., 1999) states that the TCO measurement accuracy is ±1 % for direct sun TCO. This indicates a quite good agreement between the two instruments. In any case, the negative MBD values in all periods except the last indicate some influence of the stray-light effect. However, the magnitude of the positive

MBD value in the last period, where TCO data are already corrected for the stray-light effect, suggests that this influence in the past was approximately within the range of typical instrumental error.

Table C1 Statistical parameters (RMSE, STD, MBD and R<sup>2</sup>) of the quantitative comparison of TCO between the Brewer MKIII and MKIV instruments, both located at Poprad-Gánovce, during five intercalibration periods.

| Intercalibration periods | RMSE (DU) | STD (DU) | MBD (DU) | $R^2$ |
|--------------------------|-----------|----------|----------|-------|
| 2015-06-01 - 2017-05-18  | 7.2       | 7.0      | -1.9     | 0.967 |
| 2017-05-19 - 2019-05-15  | 5.0       | 3.7      | -3.3     | 0.992 |
| 2019-05-16 - 2021-08-31  | 3.6       | 3.4      | -0.9     | 0.993 |
| 2021-09-01 - 2023-06-26  | 3.9       | 3.3      | -2.0     | 0.992 |
| 2023-06-27 - 2024-05-31  | 3.0       | 2.2      | 2.0      | 0.998 |

Given the very long measurement series, starting as early as 1993, it was also decided to analyse the possible impact of the stray-light effect using a statistical analysis method described by Savastiouk et al. (2023). The method is based on plotting a large number of individual retrievals TCO as deviations from their respective daily medians. If little or no systematic daily variations in TCO are expected, then the differences from the daily medians should be almost randomly distributed around zero. Hence, any clear departure from zero increasing with ozone slant column density (SCD) could be a sign of the stray-light effect (Savastiouk et al., 2023).

The analysis of Poprad-Gánovce data from Brewer No. 97 was carried out under the following conditions: all data were divided into 16 intercalibration periods; within each period the data were binned in intervals of 100 DU SCD; only days with a standard deviation of TCO < 3 DU were taken into account; and bins with a low number of points (< 50) were not plotted owing to insufficient statistics. Subsequently, for each intercalibration period, bins of SCD values and their corresponding relative deviations (averaged over the entire period) were linearly interpolated. The values in Table C2 were therefore obtained from the linear interpolation equation.

It is important to note that the values for 2024 already refer to the intercalibration period in which TCO data were obtained by applying a correction for the stray-light effect. TCO values for previous years could not be corrected for the stray-light effect. The deviation value for 2024 at SCD 1000 DU is -0.24 %, which is the same magnitude as the average over all 15 preceding periods. In the case of SCD 2000 DU, the mean value of -0.80 % is even lower than the 2024 value of -0.88 %. This indicates that the calibrations performed by IOS were generally able to reliably eliminate the influence of stray-light effect, with the degree of elimination reaching levels comparable to those obtained using the correction for the stray-light effect in the last period.

Savastiouk et al. (2023) report that, for a typical single monochromator Brewer, stray-light leads to an underestimation of ozone of approximately 1 % at SCD 1000 DU and can exceed 5 % at SCD 2000 DU. The average relative deviation for SCD 1000 DU (–0.24 %) over 15 intercalibration periods is approximately four times lower than the reported typical value. For SCD 2000 DU, the average relative deviation (–0.80 %) is even about six times lower. This comparison assures us that past values can also be used relatively reliably, as the effect of stray-light was largely eliminated.

Table C2 Time evolution of the percentage deviation of TCO from its respective daily median for SCD values of 1000 DU and 2000 DU over 16 intercalibration periods. The year indicates the midpoint of each two-year intercalibration period.

| SCD     | 1994  | 1996  | 1998  | 2000  | 2002  | 2004  | 2006  | 2008  |
|---------|-------|-------|-------|-------|-------|-------|-------|-------|
| 1000 DU | -0.10 | -0.20 | -0.16 | -0.27 | -0.24 | -0.15 | -0.12 | -0.19 |
| 2000 DU | -0.32 | -0.56 | -0.57 | -0.75 | -0.81 | -0.54 | -0.44 | -0.58 |
| SCD     | 2010  | 2012  | 2014  | 2016  | 2018  | 2020  | 2022  | 2024  |
| 1000 DU | -0.28 | -0.33 | -0.32 | -0.37 | -0.18 | -0.27 | -0.37 | -0.24 |
| 2000 DU | -1.05 | -1.16 | -1.10 | -1.29 | -0.70 | -0.75 | -1.30 | -0.88 |

Initially, the goal was to correct TCO for the influence of the stray-light effect using the results of statistical analysis and the known stray-light constants from the 2023 calibration. However, in the end, after further consideration, it became clear that 795 this approach was incorrect. In our case, it was decided not to apply a posteriori stray-light correction to the historical data (before calibration in 2023). The main reason is that, in earlier calibrations, the ozone absorption coefficient was often derived together with the ETC by regression against a reference instrument, in order to obtain agreement with the reference Brewer. As a consequence, the resulting absorption coefficient frequently deviated from the value determined from the dispersion test. This procedure effectively compensated, at least partly, for the stray-light effect of the calibrated instrument. 800 If one applies the recently determined stray-light constants to the historical datasets, it would also be necessary to adjust the corresponding ETC and absorption coefficient in a consistent way; otherwise, the recalculated TCO values would no longer match the original results. For this reason, applying only the new stray-light constants without simultaneously updating both ETC and the absorption coefficient would not yield reliable results. A potential re-evaluation of the historical Brewer No. 97 data series since 1993 would, in principle, be possible. However, this would require a complete recalculation of the 805 calibration of the Brewer No. 97 at Poprad-Gánovce against the reference instrument No. 17. This is therefore beyond the scope of the currently achievable possibilities, but it represents a challenge for the future.

# Appendix D: Description of hypothetical TCO calculation

In this section, the procedure followed to separate the contribution of tropopause height is described. The series of hypothetical TCO ( $TCO_{hy}_{,i}$ ) was computed by subtracting the variation in total ozone associated with changes in tropopause height ( $TCO_{trop,i}$ ) from the observed TCO measurements ( $TCO_{obs,i}$ ):

$$TCO_{hyp,i} = TCO_{obs,i} - TCO_{trop,i} . (D1)$$

To determine the  $TCO_{trop,i}$  term, the relationship between variations in TCO as a function of tropopause height has been analysed. Based on the methodology described in the manuscript (Figure 5, Table 5), this dependence was studied by distinguishing among four seasons, each consisting of three months: February/March/April, May/June/July, August/September/October and November/December/January. After classifying the data points by season, the TCO values were grouped into height intervals for each season and the corresponding mean was determined. Finally, these points were linearly fitted, as shown in Figure D1. The numerical relationship between TCO and tropopause height,  $\alpha$ , can be estimated from the slope resulting from the linear fit (Table D1). For simplicity, Figure 5 and Table 5 in the article refer only to the May/June/July and November/December/January seasons. The other two seasons have been added to Figure D1 and Table D1 below. It is important to mention that, taking into account the p-values obtained from the linear fits, trends have been found to be statistically significant for all seasons.

Once the variation of the TCO as a function of the tropopause height has been quantified, the TCO<sub>trop,i</sub> can be computed as:

$$TCO_{trop,i} = \alpha \left( H_i - \overline{H} \right),$$
 (D2)

where  $H_i$  is each measurement of the tropopause height and  $\overline{H}$  is the mean height, in such a way that variations in tropopause height will be considered with respect to this point. Equation D2 has been applied separately to the dataset of each season, so the corresponding  $\overline{H}$  is computed in each case. In particular,  $\overline{H}_{Feb/Mar/Apr} = 10.4$  km,  $\overline{H}_{MayJun/Jul} = 11.4$  km,  $\overline{H}_{Aug/Sep/Oct} = 11.7$  km and  $\overline{H}_{Nov/Dec/Jan} = 10.7$  km. Furthermore, as it has been already mentioned,  $\alpha$  can be found in Table D1 for each season. Taking into account Equations D1 and D2, the hypothetical TCO will be given by:

$$TCO_{hyp,i} = TCO_{obs,i} - \alpha (H_i - \overline{H})$$
 (D3)

Figure D1 Linear fit of TCO means obtained for different ranges of tropopause height during the May, June, July (purple); August, September, October (blue); November, December, January (red); and February, March, April (green) periods. The data set considered for the plot corresponds to days between 18 August 1993 and 31 May 2024, when daily means for both tropopause height and TCO are available.

Table D1 Parameters (slope and R<sup>2</sup>) resulting from the linear regression analysis of TCO means computed for different ranges of tropopause height, based on data from 18 August 1993 to 31 May 2024.

| Months          | Slope<br>(DU/km) | p                    | $R^2$ |
|-----------------|------------------|----------------------|-------|
| May/June/July   | $-11.5 \pm 0.6$  | $8.2 \times 10^{-9}$ | 0.98  |
| Aug/Sep/Oct     | $-6.5 \pm 0.9$   | $1.3 \times 10^{-4}$ | 0.89  |
| Nov/Dec/Jan     | $-11.7 \pm 1.0$  | $3.0 \times 10^{-6}$ | 0.94  |
| Feb/March/April | $-15.1 \pm 0.9$  | $3.0 \times 10^{-6}$ | 0.98  |

# Code availability

The Python code for calculating the AOD is available from the corresponding author upon reasonable request.

#### Data availability

The data used in this study are available from the corresponding author upon reasonable request. TCO data are also available from the WOUDC (https://woudc.org/data/stations/331; last accessed: 2 October 2025) and EUBREWNET (https://eubrewnet.aemet.es/eubrewnet/station/view/33; last accessed: 2 October 2025) databases.

#### 845 Author contributions

PH developed the initial concept of the study, calculated the AOD, and coordinated the work. MGS and VM performed the data analysis and contributed to the preparation of the manuscript. AP and VE contributed to the editing of the manuscript. JD was responsible for the acquisition of aerological measurement data. MS was in charge of the calibration of the Brewer spectrophotometer and the development of O3Brewer software. MS supported cooperation between the Czech and Slovak hydrometeorological institutes.

# **Competing interests**

The authors have no competing interests to declare.

# Acknowledgments

The publication is based upon work of COST Action HARMONIA (International network for harmonization of atmospheric aerosol retrievals from ground based photometers), CA2119, supported by COST (European Cooperation in Science and Technology). We would like to express our sincere gratitude to all current and former colleagues who contributed to ensuring the measurements at the Poprad-Gánovce station.

#### **Financial support**

This work was supported both from COST Action CA21119 Harmonia: International network for harmonisation of atmospheric aerosol retrievals from ground based photometers, supported by COST (European Cooperation in Science and Technology). Participation of M. Garcia-Suñer and V. Matos was made possible respectively by grant PREP2022-000658 from the Spanish Science and Innovation Ministry, and project PID2022-138730OB-I00, funded by the Spanish Ministry of Economy and Competitiveness and the European Regional Development Fund.

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
