# Peer review of "years of total column ozone and aerosol optical depth measurements using the Brewer spectrophotometer in Poprad-Gánovce, Slovakia"

_EGUsphere, 2025_

## Author Comment (AC1)

**Answers to comments of anonymous referee #1**

Thank you very much for your comments and suggestions. Our responses are written in blue in the text.

The manuscript presents 30 years of total column ozone and aerosol optical depth data measured by a Brewer Spectrometer at the Poprad Ganovce station in Slovakia, in Eastern Europe. In addition, tropopause height data are presented from regular radiosonde lauches at the same site. In large parts, the paper is an update of a previous paper by Hrabčák et al. (2018), which uses the same instrument and methods. The authors report a significant decreasing trend of aerosol optical depth, in all seasons, a significant increasing trend in tropopause height, throughout most of the year, and little or no trend in total column ozone.

Consistent long-term observations, like the ones present here, are important and deserve publication in a journal like ACP. While the manuscript presents no ground-braking new results, it still confirms findings of other studies, and helps with our understanding of long-term changes in the atmosphere. I suggest publication in ACP after a few generally minor revisions.

Section 2.4, in my opinion is rather lengthy, difficult to understand, and essentially a complete repeat of what is already presented in Hrabčák et al. (2018). I suggest to remove most of section 2.4, only describe the most salient points, and otherwise refer to Hrabčák et al. (2018). Essentially, to get aerosol optical depth, you need the measured intensity S from the Brewer, the ETC S_0, and you have to subtract ozone and Rayleigh optical depths times their air-masses. Why not write the relevant Equation that provides aerosol optical depth, and then say that Hrabčák et al. (2018) explain how to get all the parameters in that Equation. If there is anything different from Hrabčák et al. (2018), then explain that. Doing this will reduce Section 2.4 from about 100 lines to 10 or 20 lines, and will make the manuscript much more readable.

The revised version of the manuscript will contain a significantly reduced amount of this section. However, the goal is to move most of the text to the appendix.

Figure 2: you might want to show another panel, which would present the annual cycle of tropopause height in a similar fashion. You might be surprised how closely the annual cycle of tropopause height mirrors the annual cycle of total column ozone.

Thank you for this suggestion. Indeed, the results obtained show the same behaviour as the Reviewer had pointed out. Then, Figure 2a from the article can be substituted by Figure 1 below. In this plot, the inter-annual variation of the tropopause height, computed based on monthly means, has been included next to the annual cycle of TCO. Left vertical axis represents Tropopause height in m, while right vertical axis depicts TCO in DU. It can be noticed that both magnitudes show an opposite behaviour: mean TCO values peak on March (due to Brewer-Dobson circulation, as suggested in the article), coinciding with the minimum in tropopause height. The lowest TCO concentrations are detected in October, but the maximum in tropopause height means occurs in August. Despite the shift, the annual cycle of tropopause height is in remarkable anticorrelation with the annual cycle of TCO.

[Figure]

Figure 1 Boxplot showing the statistical distribution of the tropopause height (left vertical axis) and TCO (right vertical axis) for each month based on data from September 1993 to May 2024. The means are represented by solid points. The horizontal lines inside the boxes indicate the medians. The boxes extend from the 25th percentile (***U25***) to the 75th percentile (***U75***). Additionally, the lower and upper whiskers represent the corresponding minimum and maximum values, respectively.

Lines 286 to 292: I would drop this paragraph. It is not needed here.

The authors acknowledge that this paragraph may be overly descriptive and might slow down the pace of reading in the results section. However, they consider it to be quite relevant, as it describes the distribution and transport of ozone in the atmosphere, which is key to interpreting the results. So, these paragraphs:

"*TCO varies strongly with latitude over the globe, with the largest values occurring at middle and high latitudes during most of the year. This distribution is the result of the large-scale circulation of air in the stratosphere that slowly transports ozone rich air from high altitudes in the tropics, where ozone production from solar ultraviolet radiation is largest, toward the poles. Ozone accumulates at middle and high latitudes, increasing the vertical extent of the ozone layer and, at the same time, TCO. The TCO is generally smallest in the tropics for all seasons. An exception since the mid-1980s is the region of low values of ozone over Antarctica during spring in the Southern Hemisphere, a phenomenon known as the Antarctic ozone hole (Salawitch et al., 2023).*

*TCO also varies with season. During spring, it exhibits maxima at latitudes poleward of about 45° N in the Northern Hemisphere and between 45° and 60° S in the Southern Hemisphere. These spring maxima are a result of increased transport of ozone from its source region in the tropics toward high latitudes during late autumn and winter. This poleward ozone transport is much weaker during the summer and early autumn periods and is weaker overall in the Southern Hemisphere (Salawitch et al., 2023). This natural seasonal cycle can be clearly observed in Fig. 2a. Furthermore, it has been reported that the Brewer-Dobson circulation seems to have accelerated during the last years due to the increased presence of greenhouse gases in the atmosphere (Braesicke et al., 2003; Butchart et al., 2006). Other natural atmospheric cycles (e.g., the Quasi Biennial Oscillation, El Niño-Southern Oscillation, Arctic*

*and Antarctic Oscillations, the solar cycle, etc.) have also been found to influence TCO (Coldewey-Egbers et al., 2022). Since these cycles operate on different timescales, assessing the individual impact of each on TCO is challenging."*

can be added to the Introduction, in a new section describing the Brewer-Dobson circulation and/or other relevant processes related to TCO changes, as well as how and to which extent they affect it.

Tables 3 to 6: It would be good to have additional columns giving uncertainty estimates for the trends.

Indeed. The authors acknowledge that these tables could contain more information relevant for the study. Thereby, they have estimated the uncertainty for the trends and have assessed their significance based on the p-value obtained as output from the Ordinary Least Squares (OLS) regression algorithm in Python.

Table 3 from the manuscript corresponds to Table 1 below. In this case, the uncertainty for the trends is given by the standard error associated to the value of the slope, which is an output from the OLS algorithm. In the case of the TCO, the uncertainties are larger than the value of the slope (except for summer). This, along with the large p-values obtained (0.83, 0.24, 0.77, 0.71 and 0.91 for spring, summer, autumn, winter and annual, respectively), confirms the apparent statistical insignificance of the seasonal trend for the TCO. On the contrary, p-values obtained for the seasonal $AOD_{320}$ indicate highly significant trends: $1.5 \times 10^{-8}$, $1.7 \times 10^{-6}$, $9.4 \times 10^{-7}$, $2 \times 10^{-4}$ and $8.6 \times 10^{-1}$ for the spring, summer, autumn, winter and annual analyses, respectively. Figure 3 from the manuscript depicts these behaviours. Fig. 3b clearly shows decreasing trends, while Fig. 3a shows lines with a slight slope.

Table 2 (which is Table 4 from the manuscript) focuses on the study of $AOD_{320}$ trends for each month of the year. The estimated uncertainties for the slopes of the linear regression have been obtained from the results of the OLS model in Python, as already mentioned. Regarding the p-values, they all indicate very significant trends: 0.028, $1.7 \times 10^{-3}$, $1.1 \times 10^{-4}$, $1.6 \times 10^{-5}$, $1.5 \times 10^{-6}$, $9.8 \times 10^{-5}$, $9 \times 10^{-4}$, $1.1 \times 10^{-4}$, $4.7 \times 10^{-4}$, $1.1 \times 10^{-5}$, $1.5 \times 10^{-4}$ and 0.017 from January to December, respectively.

Besides the linear regression model, the Python algorithms for the Mann-Kendall test and Sen's slope (*pymannkendall*) were applied. The error of Sen's slope, $u_S$, has been estimated based on their upper and lower limits at the 95% confidence interval ($U_{95}$ and $L_{95}$, respectively), in such a way that $u_S = \frac{U_{95} - L_{95}}{2}$. Taking into account the slope values and their uncertainties, the compatibility between the values obtained from the linear regression and the Mann-Kendall test can be confirmed.

The results from the linear regression analysis performed to quantify the seasonal dependence of TCO in tropopause height are summarised in Table 3 (Table 5 from the manuscript). The uncertainty associated with the slope values has been added. Based on the $R^2$ and p-values ($8.2 \times 10^{-9}$ and $3.0 \times 10^{-6}$ for May/June/July and November/December/January, respectively), a statistically significant inverse relationship between TCO and tropopause height can be identified.

Finally, Table 4 (Table 6 in the manuscript) presents the results from applying the linear regression and Mann-Kendall tests to deseasonalised monthly means of tropopause height and TCO. The corresponding errors have been determined as mentioned above. For the tropopause height, the errors are higher than the slope values in January, February, March (in the case of the Mann-Kendall analysis), April, May and December. In fact, the corresponding $R^2$ and p-values ($0.50, 0.59, 0.28, 0.37, 0.70$ and $0.41$,

respectively) from the linear regression analysis indicate the insignificance of the trends. For the remaining months, high p-values are also obtained in July, October and November (0.13, 0.19, 0.10, respectively); while significant dependencies are found in June, August and September (0.04, 0.008, 0.04, respectively), in agreement with the Z values from the Mann-Kendall test representing statistically significant trends at least at a 95 % confidence level (CL).

Regarding the TCO and focusing on the p-values from the linear regression test, only the trend in August can be considered significant with 90 % confidence (p-value = 0.06). For the rest of the year, the p-values are very high (0.19, 0.73, 0.47, 0.43, 0.58, 0.80, 0.65, 0.23, 0.69, 0.34 and 0.99), representing statistically insignificant trends. These conclusions are in line with the results of the Mann-Kendall test, since, according to the Z values, no statistically significant relationship can be found with a 95 % CL in the case of the TCO. In addition, it can be observed that, in both analyses, the slope errors are generally greater than the corresponding value. The exception is August, as well as January and September in the case of the linear regression test.

Table 1 Parameters obtained from the linear regression analysis of seasonal and annual TCO and $AOD_{320}$, based on weighted means from 1994 to 2023.

| | $TCO$ | | $AOD_{320}$ | |
|---|---|---|---|---|
| | Trend (DU/decade) | $R^2$ | Trend (decade$^{-1}$) | $R^2$ |
| Spring | $0 \pm 2$ | 0.002 | $-0.074 \pm 0.009$ | 0.69 |
| Summer | $-1.6 \pm 1.3$ | 0.05 | $-0.068 \pm 0.011$ | 0.56 |
| Autumn | $-0.5 \pm 1.7$ | 0.003 | $-0.053 \pm 0.009$ | 0.58 |
| Winter | $1 \pm 3$ | 0.005 | $-0.032 \pm 0.008$ | 0.39 |
| **Annual** | $-0.2 \pm 1.4$ | 0.0005 | $-0.057 \pm 0.005$ | 0.82 |

Table 2 Parameters (trends and $R^2$) obtained from the linear regression analysis of $AOD_{320}$ for each month of the year, based on data from September 1993 to May 2024. In addition, results from the Mann-Kendall test are also included. Specifically, **Z** represents the test statistic of the Mann-Kendall test, and **S** denotes Sen's slope. **Z** values indicating statistically significant trends at the 95 % confidence level ($|Z| > 1.96$) have been highlighted in bold.

| | Linear regression | | Mann-Kendall test | |
|---|---|---|---|---|
| | Trend (decade$^{-1}$) | $R^2$ | Z | S (decade$^{-1}$) |
| **January** | $-0.021 \pm 0.009$ | 0.16 | **-2.2** | $-0.018 \pm 0.012$ |
| **February** | $-0.043 \pm 0.012$ | 0.30 | **-2.8** | $-0.044 \pm 0.019$ |
| **March** | $-0.069 \pm 0.015$ | 0.42 | **-4.2** | $-0.060 \pm 0.019$ |
| **April** | $-0.085 \pm 0.016$ | 0.49 | **-3.8** | $-0.09 \pm 0.03$ |
| **May** | $-0.067 \pm 0.011$ | 0.57 | **-4.6** | $-0.072 \pm 0.013$ |
| **June** | $-0.059 \pm 0.013$ | 0.42 | **-3.2** | $-0.05 \pm 0.02$ |
| **July** | $-0.062 \pm 0.017$ | 0.33 | **-3.1** | $-0.06 \pm 0.02$ |
| **August** | $-0.084 \pm 0.019$ | 0.42 | **-3.4** | $-0.08 \pm 0.03$ |
| **September** | $-0.070 \pm 0.018$ | 0.36 | **-3.6** | $-0.07 \pm 0.02$ |
| **October** | $-0.052 \pm 0.010$ | 0.50 | **-4.2** | $-0.049 \pm 0.011$ |
| **November** | $-0.039 \pm 0.009$ | 0.41 | **-3.1** | $-0.036 \pm 0.014$ |
| **December** | $-0.032 \pm 0.013$ | 0.19 | **-2.1** | $-0.023 \pm 0.014$ |

Table 3 Parameters resulting from the linear regression analysis of TCO means computed for different ranges of tropopause height, based on data from 18 August 1993 to 31 May 2024.

| Months | Slope (DU/km) | $R^2$ |
|---|---|---|
| May/June/July | -11.5 ± 0.6 | 0.98 |
| Nov/Dec/Jan | -11.7 ± 1.0 | 0.94 |

Table 4 Parameters (trend and $R^2$) obtained from the linear regression analysis of tropopause height and TCO for each month of the year, based on data from January 1994 to December 2023. Additionally, results from the Mann-Kendall test are included. Specifically, $Z$ represents the Mann-Kendall test statistic, and $S$ denotes Sen's slope. $Z$ values indicating statistically significant trends at a confidence level of at least 95 % ($|Z| > 1.96$) are highlighted in bold.

| | Tropopause height | | | | TCO | | | |
|---|---|---|---|---|---|---|---|---|
| | Linear regression | | Mann-Kendall test | | Linear regression | | Mann-Kendall test | |
| Month | Trend (m/decade) | $R^2$ | Z | S (m/decade) | Trend (DU/decade) | $R^2$ | Z | S (DU/decade) |
| January | -60 ± 90 | 0.016 | -1.0 | -110 ± 140 | 5 ± 3 | 0.06 | 1.2 | 4 ± 5 |
| February | 70 ± 120 | 0.011 | 0.8 | 100 ± 140 | -2 ± 5 | 0.004 | -0.5 | -4 ± 7 |
| March | 90 ± 80 | 0.04 | 1.0 | 90 ± 130 | 2 ± 3 | 0.018 | 0.6 | 2 ± 5 |
| April | 80 ± 90 | 0.03 | 0.8 | 100 ± 120 | -3 ± 3 | 0.02 | -0.9 | -3 ± 4 |
| May | -30 ± 80 | 0.005 | -0.4 | -20 ± 100 | 1 ± 2 | 0.011 | 1.2 | 3 ± 3 |
| June | 130 ± 60 | 0.14 | **2.2** | 150 ± 80 | -0.4 ± 1.7 | 0.002 | -0.5 | -1 ± 3 |
| July | 120 ± 80 | 0.08 | 1.6 | 200 ± 100 | -0.7 ± 1.6 | 0.008 | -0.5 | -1 ± 2 |
| August | 200 ± 70 | 0.23 | **2.9** | 200 ± 100 | -3.6 ± 1.8 | 0.12 | -1.8 | -4 ± 2 |
| September | 200 ± 100 | 0.14 | **2.1** | 250 ± 150 | -3 ± 2 | 0.05 | -1.4 | -3 ± 3 |
| October | 100 ± 100 | 0.06 | 1.5 | 200 ± 130 | -0.8 ± 1.9 | 0.006 | 0.1 | 0 ± 3 |
| November | 150 ± 90 | 0.09 | 1.9 | 200 ± 100 | 2 ± 2 | 0.03 | 1.0 | 2 ± 3 |
| December | 90 ± 100 | 0.02 | 0.7 | 70 ± 130 | 0 ± 3 | $5 \cdot 10^{-6}$ | -0.2 | 0 ± 4 |

Figure 6 and Table 6: It would be very interesting to see hypothetical TCO time series and trends, in which the -11.6 DU/km "dependence" on tropopause height has been backed out. Such a hypothetical time series in Fig. 6 might show a TCO increase, and the hypothetical effect of tropopause height changes. In Table 6, the hypothetical TCO trends would mostly become more positive by around 1 DU / decade. In fact, an additional Figure showing the seasonal variation of TCO trend, tropopause height related TCO trend and "hypothetical" TCO trend would be interesting. I suggest that the authors add such a Figure and discuss it. The slightly positive "hypothetical" TCO trend would be inline with ozone increases expected to to declining ODS (possibly enhanced by stronger Brewer Dobson Circulation). The discussion would give more meaning and context for tropopause height / climate change influences on total column ozone, and would round the paper nicely.

Thank you for your comment. The authors agree that the suggested figure could be a good addition to the article. The procedure followed to separate the contribution from the tropopause height is thus described below.

The series of hypothetical TCO has been computed by subtracting the variation in total ozone related to changes in the height of the tropopause, $TCO_{trop,i}$, from the TCO measurements, $TCO_{obs,i}$:

$$TCO_{hyp,i} = TCO_{obs,i} - TCO_{trop,i}. \qquad (1)$$

To determine the $TCO_{trop,i}$ term, the relationship between variations in TCO as a function of tropopause height has been analysed. Based on the methodology described in the manuscript (Figure 5, Table 5), this dependence was studied by distinguishing among four seasons, each consisting of three months: February/March/April, May/June/July, August/September/October and November/December/January. After classifying the data points by season, the TCO values were grouped into height intervals for each season and the corresponding mean was determined. Finally, these points were linearly fitted, as shown in Figure 2. The numerical relationship between total ozone and tropopause height, $\alpha$, can be estimated from the slope resulting from the linear fit (Table 5).

For simplicity, Figure 5 and Table 5 in the article refer only to the May/June/July and November/December/January seasons. The other two seasons have been added to Figure 2 and Table 5 below. It is important to mention that, taking into account the p-values obtained from the linear fits, trends have been found to be statistically significant for all seasons.

[Figure]

Figure 2 Linear fit of TCO means obtained for different ranges of tropopause height during the May, June, and July (purple); August, September and October (blue); November, December, and January (red); and February, March, April (green) periods. The data set considered for the plot corresponds to days between 18 August 1993 and 31 May 2024, when daily means for both tropopause height and TCO are available.

Table 5 Parameters resulting from the linear regression analysis of TCO means computed for different ranges of tropopause height, based on data from 18 August 1993 to 31 May 2024.

| Months | Slope (DU/km) | $R^2$ |
|---|---|---|
| May/June/July | $-11.5 \pm 0.6$ | 0.98 |
| Aug/Sep/Oct | $-6.5 \pm 0.9$ | 0.89 |
| Nov/Dec/Jan | $-11.7 \pm 1.0$ | 0.94 |
| Feb/March/April | $-15.1 \pm 0.9$ | 0.98 |

Once the variation of the TCO as a function of the tropopause height has been quantified, the $TCO_{trop,i}$ can be computed as:

$$TCO_{trop,i} = \alpha \, (H_i - \bar{H}) \, , \qquad (2)$$

where $H_i$ is each measurement of the tropopause height and $\bar{H}$ is the mean height, in such a way that variations in tropopause height will be considered with respect to this point. Equation 2 has been applied separately to the dataset of each season, so the corresponding $\bar{H}$ is computed in each case. In particular, $\bar{H}_{Feb/Mar/Apr} = 10.4$ km, $\bar{H}_{MayJun/Jul} = 11.4$ km, $\bar{H}_{Aug/Sep/Oct} = 11.7$ km and $\bar{H}_{Nov/Dec/Jan} = 10.7$ km. Furthermore, as it has been already mentioned, $\alpha$ can be found in Table 5 for each season. Taking into account Equations 1 and 2, the hypothetical TCO will be given by

$$TCO_{hyp,i} = TCO_{obs,i} - \alpha \, (H_i - \bar{H}) \, . \qquad (3)$$

The hypothetical TCO time series obtained has been deseasonalised and smoothed as described in the manuscript. This is shown in Figure 3a. The dashed line in the plot represents the linear fit of the data, the slope being slightly positive $(1.3 \pm 0.3\ DU/decade)$, as predicted by the Reviewer. For completeness, Figure 3b depicts the analogous temporal evolution of TCO attributed to changes in tropopause height. As expected, the slope is negative $(-1.30 \pm 0.10\ DU/decade)$. It should also be noted that it is similar in absolute value to the slope obtained for the hypothetical TCO, to which factors such as natural atmospheric cycles, next climate change influences, and the presence of ODS contribute.

The temporal evolution of the hypothetical and tropopause TCO series has also been approached by analysing their trends over the years for each month, analogous to the manuscript. Both linear regression and Mann-Kendall's tests + Sen's slope have been applied. The results are summarised in Table 6. When comparing the monthly trends of the TCO (Table 4) with those of the hypothetical TCO (Table 6), a weakly increasing trend is evident in the former only in January, whereas in the latter it also appears in March and November. It is noteworthy that not all of the decreasing trend in TCO in August can be explained solely by the tropopause increase, as the $TCO_{hypo}$ trend for this month remains negative. Another reason for the decrease in TCO in August could be related to changes in large-scale circulation patterns in the stratosphere.

[Figure]

Figure 3 Representation of the time evolution of deseasonalised monthly running means of hypothetical TCO (a) and TCO related to the height of the tropopause (b) from March 1994 to November 2023 (circles). Dashed lines represent linear fits to the data.

Table 6 Parameters (trend and R2) obtained from the linear regression analysis of hypothetical and tropopause TCO for each month of the year, based on data from January 1994 to December 2023. Additionally, results from the Mann-Kendall test are included. Specifically, $Z$ represents the Mann-Kendall test statistic, and $S$ denotes Sen's slope. $Z$ values indicating statistically significant trends at a confidence level of at least 95 % ($|Z| > 1.96$) are highlighted in bold.

| | $TCO_{hypo}$ | | | | $TCO_{tropo}$ | | | |
| | Linear regression | | Mann-Kendall test | | Linear regression | | Mann-Kendall test | |
| Month | Trend (DU/decade) | $R^2$ | Z | S (DU/decade) | Trend (DU/decade) | $R^2$ | Z | S (DU/decade) |
|---|---|---|---|---|---|---|---|---|
| January | 4 ± 3 | 0.06 | 1.2 | 4 ± 4 | 0.6 ± 1.1 | 0.009 | 0.4 | 0.5 ± 1.8 |
| February | 0 ± 4 | 0.0004 | -0.2 | -1 ± 5 | -1 ± 2 | 0.016 | -0.9 | -2 ± 3 |
| March | 4 ± 3 | 0.08 | 1.6 | 5 ± 3 | -2.0 ± 1.5 | 0.06 | -1.2 | -2 ± 2 |
| April | -1 ± 3 | 0.007 | -0.6 | -1 ± 4 | -1.4 ± 1.5 | 0.03 | -0.9 | -2 ± 2 |
| May | 1 ± 2 | 0.004 | 0.6 | 2 ± 3 | 0.7 ± 0.9 | 0.019 | 0.8 | 0.9 ± 1.3 |
| June | 1.2 ± 1.7 | 0.018 | 0.5 | 1 ± 3 | -1.7 ± 0.7 | 0.17 | **-1.96** | -1.5 ± 1.1 |
| July | 0.1 ± 1.5 | 4·10⁻⁵ | -0.4 | -0.4 ± 1.7 | -1.1 ± 0.9 | 0.05 | -1.14 | -1.2 ± 1.3 |
| August | -2.7 ± 1.7 | 0.08 | -1.5 | -3 ± 2 | -1.3 ± 0.5 | 0.21 | **-2.5** | -1.4 ± 0.6 |
| September | -1.2 ± 1.8 | 0.015 | -1.0 | -1 ± 2 | -1.6 ± 0.7 | 0.17 | **-2.7** | -2.0 ± 0.9 |
| October | 0.5 ± 1.5 | 0.004 | 0.6 | 1 ± 2 | -1.3 ± 0.7 | 0.12 | -1.8 | -1.4 ± 1.0 |
| November | 4.5 ± 1.7 | 0.20 | 1.93 | 4 ± 2 | -2.3 ± 1.2 | 0.12 | **-2.1** | -2.3 ± 1.6 |
| December | 2 ± 2 | 0.017 | 0.7 | 2 ± 3 | -1.7 ± 1.6 | 0.04 | -1.0 | -1.6 ± 1.8 |

When comparing the results of both tests, an agreement between the values of the linear regression slopes and Sen's slopes can be noticed. Furthermore, it is important to mention that no statistically significant trend has been found except in November, where the corresponding p-value in the linear regression analysis is 0.015. For the other months, p-values are quite high: 0.19, 0.92, 0.14, 0.66, 0.74, 0.48, 0.97, 0.12, 0.52, 0.66 and 0.50 for January to October and December, respectively).

Regarding $TCO_{tropo}$, the trends are negative or close to 0, as expected, showing the strong correlation between the decrease in TCO and the increase in tropopause height. In this case, statistically significant trends are observed in June, August, September, October (for the linear regression analysis) and November, with corresponding p-values from the linear regression analysis of 0.025, 0.011, 0.023, 0.059 and 0.063. For the other months, these values are high: 0.63, 0.51, 0.19, 0.34, 0.47, 0.26 and 0.28 for January to May, July and December, respectively.

Finally, Figure 4 below is the one suggested by the Reviewer. This plot clearly illustrates what has been observed throughout this discussion. The hypothetical TCO shows a positive evolution over time, while the $TCO_{tropo}$ trend is negative. When combined, the overall trend is slightly positive, but statistically insignificant. Therefore, it can be concluded that the TCO in the atmosphere is governed by two main components: a decrease associated with the rising tropopause height (Figure 6b), represented by $TCO_{tropo}$, and an increase represented by $TCO_{hypo}$. The second factor can be attributed, on the one hand, to the implementation of policies aimed at reducing ODS emissions. On the other hand, the acceleration of the Brewer–Dobson circulation due to climate change probably contributes to the increase in TCO by enhancing the transport of ozone to mid-latitude sites such as Poprad-Gánovce.

[Figure]

Figure 4 Representation of the time evolution of deseasonalised monthly running means of hypothetical (fuchsia), tropopause (blue) and observed (green) TCO from March 1994 to November 2023. Dashed lines represent linear fits to the data.

The authors consider this discussion to be of great interest, and all or a part of it will be added to the article. Furthermore, Figure 4 and Table 6 will be added to the manuscript to support the discussion.

---

## Author Comment (AC2)

**Answers to comments of anonymous referee #2**

Thank you very much for your comments and suggestions. Our responses are written in blue in the text.

General Comments:

This study presents a valuable long-term dataset of total ozone and aerosol optical depth (AOD) derived from Brewer spectrophotometer observations. The extended temporal coverage makes the work particularly relevant. Its scientific impact would be significantly enhanced if the dataset were made publicly available through established repositories such as the World Ozone and Ultraviolet Radiation Data Centre (WOUDC) and/or EUBREWNET, and ideally registered with a DOI to ensure long-term accessibility and citation.

Related to World Ozone and Ultraviolet Radiation Data Centre (WOUDC), total ozone data (since 1993) for the Poprad-Gánovce station are already part of this international database. Reports are sent on a monthly basis. The raw data from Brewer 097 are sent to the EUBREWNET network multiple times per day (near real time), available since 2014. EUBREWNET then processes and provides TCO data. However, data submission to the World Data Center PANGAEA is also being considered.

The AOD retrieval methodology, originally described by Hrabčák (2018), is extensively detailed in the manuscript. I recommend condensing this section and referring to the original publication, focusing instead on the specific updates or modifications introduced in the current study.

The revised version of the manuscript will contain a significantly reduced amount of this section. However, the goal is to move most of the text to the appendix.

In contrast, the description of the total ozone retrieval process is relatively brief. It would be beneficial to expand this section to include details on the o3brewer software setup, including the application of the Standard Lamp correction. Calibration procedures, and how major repairs or maintenance events were handled is also valuable. Clarifying how calibration changes were applied retrospectively and outlining the traceability of Brewer #97 to the reference triad would improve the technical transparency of the study. Additionally, how does the IOS traveling standard compare with the reference triad over the 30-year period? This comparison is essential for assessing long-term consistency.

In view of the aforementioned proposals, two subchapters to which the proposals relate have been revised and expanded. The edited version looks like this:

**2.2 Brewer ozone spectrophotometer**

[revised manuscript text omitted]

I suggest to include the comparison of the presented dataset with existing observations archived in WOUDC (Station 331, (https://woudc.org/data/stations/331) and EUBREWNET (https://eubrewnet.aemet.es/eubrewnet/station/view/33).

Indeed. The authors agree with the Referee on the importance of assessing the compatibility of the TCO datasets provided as a product of different networks, as well as evaluating their compatibility with the dataset exploited in this study. After all, consistency between datasets will be essential to ensure that data processing has been carried out successfully. Hence, two basic comparisons have been performed, both based on monthly means. The first is a naive approach based on plotting the temporal evolution of the monthly TCO means obtained for each dataset, so that they can be compared visually (Figure 1). It is important to note that the EUBREWNET Lv 1.5 dataset dates back to 2013, much later than those used in this study and those from WOUDC.

The TCO data for the Poprad-Gánovce station in WOUDC are produced using the mentioned O3Brewer software. This is the same software that was used to calculate the total ozone data for the article under review. In particular, older data in WOUDC were calculated using previous versions of the software, and this may lead to small discrepancies, as the software has been continuously developed over the years. Further, it is important to note that the data submitted to WOUDC once per month for the preceding month also include daily averages derived solely from the calculation of TCO using measurements of diffuse solar radiation (so-called ZS measurements). This applies to days when it was not possible to retrieve TCO from DS measurements. ZS measurements are subject to considerable uncertainty. In the article under review, the TCO dataset is derived exclusively from DS measurements. A comparison of these two different data sets may therefore lead to discrepancies, particularly during months with a higher proportion of ZS measurements (November–February).

At first glance, it can be said that there is a general agreement between the datasets. However, some discrepancies can be identified at the extremes (maxima and minima) between the calculated monthly averages and those from WOUDC. Something similar is found when comparing the EUBREWNET TCO means with those of the other two datasets, as EUBREWNET values tend to be higher at the maxima and lower in the minima.

[Figure]

Figure 1 Evolution of TCO monthly means over time obtained from measurements by the Brewer ozone spectrophotometer in Poprad-Gánovce. Blue triangles correspond to values processed by the World Ozone and Ultraviolet Radiation Data Centre (WOUDC), purple squares are products of the European Brewer Network (EUBREWNET), and golden circles are the TCO values determined for this study. The WOUDC product (as well as the dataset used in this study) covers the period from August 1993 to May 2024, while EUBREWNET Lv 1.5 TCO values are available from August 2013.

The other analysis consists of evaluating the alignment of the data with respect to the line $y = x$ when plotting the TCO means corresponding to the same year and month (Figure 2). Furthermore, in order to quantitatively study the discrepancies, some statistical parameters have been computed: root mean square error (RMSE), std of the differences, mean bias deviation (MBD) and $R^2$. These are summarised in Table 1. In general, based on Figure 2, it can be stated that the three data sets are compatible, as the differences between them are relatively small.

[Figure]

Figure 2 Scatter plots for the comparison between monthly TCO means provided by different datasets: (a) WOUDC and those determined for the analysis in the manuscript; (b) EUBREWNET and TCO for this study; and (c) EUBREWNET and WOUDC. The green dashed line indicates the plot $y = x$.

The discrepancies between WOUDC and the calculated TCO are the smallest, with an RMSE and standard deviation (STD) of the differences of 5.00 DU and 3.42 DU, respectively. In the comparison between EUBREWNET and the calculated TCO, these values are slightly higher (5.13 DU and 3.66 DU, respectively), but much lower than the values corresponding to EUBREWNET/WOUDC, 7.26 DU and 4.67 DU, respectively. However, it should be noted that, in the case of the latter two, the number of points compared is much smaller, as the EUBREWNET dataset begins in 2013. These statistical parameters measure the discrepancies between the compared values, showing less dispersion in the case of the comparisons involving the dataset used in this analysis. The same trend is observed with the MBD, which is -1.90 DU for WOUDC/Calc., 2.45 DU for EUBREWNET/Calc. and 4.48 DU for EUBREWNET/WOUDC. The negative sign for WOUDC/Calc. indicates that mean TCO for WOUDC are slightly higher than those calculated. Following the same reasoning, the calculated TCO are slightly higher than those of EUBREWNET, and the WOUDC values are consistently higher than those of EUBREWNET. All these parameters are much smaller than the measured TCO (~200-500) confirming

the compatibility among the values of the different datasets. Finally, the R²≈1 values in each case indicate the goodness of the fits, supporting the claim regarding the agreement between datasets.

As mentioned above, the authors consider this comparison quite relevant to the study. Therefore, they consider that all or a part of this discussion can be added to the appendix of manuscript.

Table 1 Statistical parameters for the quantitative study comparing the TCO of the WOUDC and EUBREWNET networks, as well as the TCO values used in this study.

|  | RMSE (DU) | $STD$ (DU) | MBD (DU) | $R^2$ |
|---|---|---|---|---|
| WOUDC/Calc. | 5.00 | 3.42 | -1.90 | 0.98 |
| EUBREWNET/Calc. | 5.13 | 3.66 | 2.45 | 0.98 |
| EUBREWNET/WOUDC | 7.26 | 4.67 | 4.48 | 0.95 |

Note: We have not found any WOUDC or EUBREWNET AOD datasets to compare.

Single-monochromator Brewer spectrophotometers are subject to straylight interference, which introduces systematic biases in ozone and sulfur dioxide retrievals. This effect arises from the intrusion of longer-wavelength photons during short-wavelength measurements, leading to underestimation of trace gas concentrations (Savastiouk et al., 2023; Karppinen et al., 2014; Rimmer et al., 2018). To mitigate this, the dataset was filtered to include only observations with an airmass less than 4. However, as straylight effects scale with the ozone slant column (total column ozone × airmass), filtering based solely on airmass may be insufficient.

A double-monochromator Brewer spectrophotometer, which is not affected by straylight, has been operational at the station since 2015 (EUBREWNET Station 225). Comparative analysis between the single and double Brewer time series offers a means to validate the filtering approach. Moreover, straylight correction algorithms developed within EUBREWNET (Redondas et al., 2018) and by IOS (Savastiouk et al., 2023) can be use on the single brewer. Applying these corrections and assessing their impact on long-term ozone trends using the double Brewer as a reference could substantially improve the reliability of the dataset and enhance the interpretability of observed atmospheric changes.

We thank the Referee for drawing our attention to the issue of the stray-light effect in relation to the measurements of the Brewer spectrophotometer single monochromator, as well as for the recommended literature. At the outset, it is important to note 
[revised manuscript text omitted]

Finally, we note that we are considering including this analyses in our article, with the key points likely to appear in the main text and additional details provided in the appendix.

Concernign the analysis of the series, i reconnice the difficulty to deal with a no signifciant ozone trend, could be more appropiae to use the Multiple Linear Regresion Methods like developed in LOTUS project (GitHub - usask-arg/lotus-regression) to assest the the influence of the tropopause height.

The authors greatly appreciate the Reviewer's suggestion to approach this analysis using the LOTUS regression methodology. Given that the amount of TCO in the atmosphere depends on various complex factors, a methodology based on assessing the relative influence of each factor on the temporal evolution of TCO may be promising for addressing this problem. LOTUS regression is particularly well suited to this case, as is has been designed to study the temporal evolution of atmospheric parameters. Thus, the factors considered by the model, called "predictors", characterise the atmospheric components and processes that can trigger changes in the levels of the parameter under study. In this case, the authors have worked with the basic set of predictors from pred_baseline_pwlt.

These are ENSO (related to the "El Niño" and "La Niña" ocean oscillations), SOLAR (which characterise the solar cycle), QBOA and QBOB (representing orthogonal components of the Quasi Biennial Oscillation), AOD (a model of stratospheric AOD, which takes into account phenomena such as volcanic eruptions or massive fires), linear_pre and linear_post (parameters representing long-term linear evolutions before and after 1997, respectively. The reason for choosing 1997 is that this is when ODS levels in the atmosphere reached their maxima and began to decline as a result of policies derived from the Montreal Protocol. The establishment of this year as a turning point is arbitrary and can be changed as appropriate, although this is not the case in this study); and K (constant, a predictor with no physical meaning that is required by the model). However, another predictor, HEIGHT, has been added to take into account the influence of the height of the tropopause, given the relevance of this parameter for TCO levels.

However, before applying the model, we had to make some adjustments. Since the series characterising the predictors begins in January 1979 and runs until February 2024, while our dataset ranges from September 1993 to May 2024, we had to limit the working time interval from September 1993 to February 2024. This meant rescaling the predictors, as these are constructed so that the mean and standard deviation of the series are 0 and 1, respectively. Thus, we computed the mean $\bar{x}$ and the std $\sigma(x)$ of the predictor series in pred_baseline_pwlt, considering only the data from September 1997 to February 2024 and determined the rescaled values of the predictor for each month, $x_i'$, as $x_i' = \frac{x_i - \bar{x}}{\sigma(x)}$, where $x_i$ is the value associated with the predictor $x$ in month $i$. In this way, the mean and std of the rescaled values are 0 and 1, respectively. In the case of the HEIGHT predictor, we started with a series of monthly means of tropopause height values between September 1997 and May 2024.

Hence, the corresponding mean and std did not verify the aforementioned condition, so the same relationship was applied as to the other predictors in order to rescale the values. The importance of rescaling parameters lies in comparing the strength of each parameter. In fact, by rescaling all these predictors, they can be compared directly, allowing us to draw conclusions about which of them has the greatest influence on the evolution of the parameter. As of the TCO, we decided to work with monthly means, but introducing a set of predictors that take seasonal components into account, based on Fourier series (see this example).

The results of the analysis are summarised in Table 4. Four predictors show a highly statistical significant trend: ENSO, QBOB, Linear_post and HEIGHT (we will not take K into account, given its non-physical meaning), the latter having the greatest influence on the TCO. In fact, the trend is negative, as expected: a decrease in TCO may be related to an increase in the height of the tropopause. The effect of the B component of the QBO, which according to Wang et al. 2022 is zonally asymmetric, also seems to translate into a decrease in TCO. Conversely, a smaller -in absolute value- but positive slope for Linear_post could be interpreted as the existence of an increasing trend in TCO since 1997, which would be consistent with the reduction in ODS emissions into the atmosphere. The effect of ENSO on TCO

appears to be positive, i.e. contributing to an increase in TCO. This has also been confirmed by other studies (e.g. Zhang et al. 2015, Li et al. 2024). Therefore, the significant role of natural atmospheric cycles in assessing the temporal evolution of TCO has been demonstrated, as well as the notable influence of the tropopause height on this parameter.

The authors consider the conclusions drawn in this analysis to be quite interesting and relevant to the article. For this reason, they believe that they can be added to the manuscript. Moreover, these results can be supported by the analysis suggested by Reviewer 1, which consists of eliminating the contribution of tropopause height to changes in TCO when studying the temporal evolution and observing the TCO time series without this contribution, resulting in an increasing trend in TCO, in line with the positive slopes obtained here in relation to ENSO and Linear_post.

Table 4 Results of the LOTUS regression analysis applied to the set of monthly TCO means from measurements taken at the Poprad-Gánovce station covering from September 1993 to February 2024. The analysis yielded a $R^2 = 0.88$. The slope value is an indicator of the weight of each parameter in the model, while the sign determines its effect on the TCO (positive causes an increase and negative a decrease). CI represents the confidence interval with a confidence level of 95%. Finally, the p-values indicate the significance of the trend. Those with p < 0.01 (highlighted in orange in the table) show very significant trends.

| Predictor | Slope (unit$^{-1}$) | CI | p-value |
|---|---|---|---|
| **ENSO** | **1.8 ± 0.7** | **[0.5, 3.2]** | **0.007** |
| SOLAR | 0.7 ± 0.7 | [-0.7, 2.0] | 0.341 |
| QBOA | -0.8 ± 0.7 | [-2.1, 0.6] | 0.264 |
| **QBOB** | **-3.1 ± 0.7** | **[-4.4, -1.7]** | **1.3·10$^{-5}$** |
| AOD | 0.4 ± 1.4 | [-2.2, 3.1] | 0.758 |
| Linear_pre | 10 ± 20 | [-30, 60] | 0.515 |
| **Linear_post** | **2.0 ± 1.2** | **[-0.4, 4.3]** | **0.099** |
| **HEIGHT** | **-13.5 ± 0.8** | **[-15.2, -11.9]** | **3.5·10$^{-43}$** |
| **K** | **323.6 ± 1.9** | **[319.9, 327.4]** | **0** |


Specific comments:

I suggest to Include on Figure 1 and 2 also the series of tropopause heights

Thank you for this suggestion. Figure 3 depicts daily means of tropopause height along with daily means of TCO values. Due to the great amount of data points, the opposite behaviour that both parameters show cannot be properly distinguished. For this reason, the authors have decided to not modify Figure 1 and substitute Figure 2a by Figure 4 below. In this plot, the inter-annual variation of the tropopause height, computed based on monthly means, has been included next to the annual cycle of TCO. Left vertical axis represents Tropopause height in m, while right vertical axis depicts TCO in DU. It can be noticed that both magnitudes show an opposite behaviour: TCO values peak on March (due to Brewer-Dobson circulation, as suggested in the article), coinciding with the minimum in tropopause height. The lowest TCO concentrations are detected in October, but the maximum in tropopause height occurs in August. Despite the shift, the annual cycle of tropopause height is in remarkable anticorrelation with the annual cycle of TCO.

[Figure]

Figure 3 Daily mean values of tropopause height (left axis) and TCO (right axis) derived from measurements taken by the Brewer ozone spectrophotometer at Poprad-Gánovce from 18 August 1993 to 31 May 2024.

[Figure]

Figure 4 Boxplot showing the statistical distribution of the tropopause height (left vertical axis) and TCO (right vertical axis) for each month based on data from September 1993 to May 2024. The means are represented by solid points. The horizontal lines inside the boxes indicate the medians. The boxes extend from the 25th percentile (**U25**) to the 75th percentile (**U75**). Additionally, the lower and upper whiskers represent the corresponding minimum and maximum values, respectively.

L 97  -Homogeneus, please justify

The measurements can be regarded as homogeneous. Occasional short interruptions occurred for technical reasons, but no extended gaps such as entire months are present.

L135 Why the SO2 measurements are not reliable, please justify

The contribution of sulfur dioxide was neglected mainly due to its low impact (Arola and Koskela, 2004) and the inaccurate determination at the Poprad-Gánovce station. This site is generally considered rural with low anthropogenic influence, and $SO_2$ concentrations are therefore often close to the detection limit. In addition, the O3Brewer software settings for Poprad-Gánovce are not well optimised for the reliable retrieval of such low $SO_2$ values.

L190 how the filter attenuation are obtained

There are five ND filters and five wavelengths, resulting in a total of 25 required attenuation values. The attenuation values for the filters are determined during the instrument's calibration. The attenuation values are derived from the Filterwheel #2 Test and subsequently recorded in the FW2TEST file.

L 210 : which are the differences ?

Main differences compared to Hrabčák (2018):

1. Condition 2 (AOD limit at 320 nm):
   o Hrabčák (2018): AOD < 0.5
   o This study: AOD < 0.4
2. Condition 6 (standard deviation of TCO):
   o Hrabčák (2018): < 2.5 DU
   o This study: < 3 DU
3. Condition 7 (standard deviation of AOD):
   o Hrabčák (2018): < 0.07
   o This study: < 0.04
4. Condition 9 (determination coefficient of linear interpolation):
   o Hrabčák (2018): > 0.98
   o This study: > 0.99

In the manuscript, we plan to include the following explanation: Compared to the procedure described in Hrabčák (2018), several modifications were introduced: condition 2: the AOD threshold at 320 nm was reduced from 0.5 to 0.4, condition 6: the standard deviation of the daily TCO was relaxed from < 2.5 DU to < 3 DU, condition 7: the standard deviation of the daily AOD was tightened from < 0.07 to < 0.04, and condition 9: the required determination coefficient of the linear interpolation was increased from > 0.98 to > 0.99.

L230 : Not clear

In the end, the following criterion was applied to all determined ETCs within the given intercalibration period:

10. $$\frac{|ETC - AVERAGE(ETCs)|}{STDEV(ETCs)} < 1.5 \,,$$

where $AVERAGE(ETCs)$ is the average of the determined ETCs and $STDEV(ETCs)$ is the standard deviation. The threshold value of 1.5 in the 10th criterion was established in a manner analogous to that of the 8th criterion, i.e., it was selected based on the results of an optimization procedure. The objective was to exclude outlier values while ensuring a sufficient number of ETCs for the calculation of the final average.

L425: Table 5, include the errors

The authors greatly appreciate this suggestion and acknowledge the need to include this information when presenting slope values. For this reason, Tables 3 to 6 (corresponding to Tables 5 to 8 below) have been completed with the slope uncertainties.

Table 7 summarises the results from the linear regression analysis performed to quantify the seasonal dependence of TCO in tropopause height. Based on the $R^2$ and p-values ($8.2 \times 10^{-9}$ and $3.0 \times 10^{-6}$

for May/June/July and November/December/January, respectively), a statistically significant inverse relationship between TCO and tropopause height can be identified.

Table 5 Parameters obtained from the linear regression analysis of seasonal and annual TCO and $AOD_{320}$, based on weighted means from 1994 to 2023.

|  | $TCO$ |  | $AOD_{320}$ |  |
|---|---|---|---|---|
|  | Trend (DU/decade) | $R^2$ | Trend (decade$^{-1}$) | $R^2$ |
| Spring | $0 \pm 2$ | 0.002 | $-0.074 \pm 0.009$ | 0.69 |
| Summer | $-1.6 \pm 1.3$ | 0.05 | $-0.068 \pm 0.011$ | 0.56 |
| Autumn | $-0.5 \pm 1.7$ | 0.003 | $-0.053 \pm 0.009$ | 0.58 |
| Winter | $1 \pm 3$ | 0.005 | $-0.032 \pm 0.008$ | 0.39 |
| **Annual** | $-0.2 \pm 1.4$ | 0.0005 | $-0.057 \pm 0.005$ | 0.82 |

Table 6 Parameters (trends and $R^2$) obtained from the linear regression analysis of $AOD_{320}$ for each month of the year, based on data from September 1993 to May 2024. In addition, results from the Mann-Kendall test are also included. Specifically, **Z** represents the test statistic of the Mann-Kendall test, and **S** denotes Sen's slope. **Z** values indicating statistically significant trends at the 95 % confidence level ($|Z| > 1.96$) have been highlighted in bold.

|  | Linear regression |  | Mann-Kendall test |  |
|---|---|---|---|---|
|  | Trend (decade$^{-1}$) | $R^2$ | Z | S (decade$^{-1}$) |
| **January** | $-0.021 \pm 0.009$ | 0.16 | **-2.2** | $-0.018 \pm 0.012$ |
| **February** | $-0.043 \pm 0.012$ | 0.30 | **-2.8** | $-0.044 \pm 0.019$ |
| **March** | $-0.069 \pm 0.015$ | 0.42 | **-4.2** | $-0.060 \pm 0.019$ |
| **April** | $-0.085 \pm 0.016$ | 0.49 | **-3.8** | $-0.09 \pm 0.03$ |
| **May** | $-0.067 \pm 0.011$ | 0.57 | **-4.6** | $-0.072 \pm 0.013$ |
| **June** | $-0.059 \pm 0.013$ | 0.42 | **-3.2** | $-0.05 \pm 0.02$ |
| **July** | $-0.062 \pm 0.017$ | 0.33 | **-3.1** | $-0.06 \pm 0.02$ |
| **August** | $-0.084 \pm 0.019$ | 0.42 | **-3.4** | $-0.08 \pm 0.03$ |
| **September** | $-0.070 \pm 0.018$ | 0.36 | **-3.6** | $-0.07 \pm 0.02$ |
| **October** | $-0.052 \pm 0.010$ | 0.50 | **-4.2** | $-0.049 \pm 0.011$ |
| **November** | $-0.039 \pm 0.009$ | 0.41 | **-3.1** | $-0.036 \pm 0.014$ |
| **December** | $-0.032 \pm 0.013$ | 0.19 | **-2.1** | $-0.023 \pm 0.014$ |

Table 7 Parameters resulting from the linear regression analysis of TCO means computed for different ranges of tropopause height, based on data from 18 August 1993 to 31 May 2024.

| Months | Slope (DU/km) | $R^2$ |
|---|---|---|
| May/June/July | $-11.5 \pm 0.6$ | 0.98 |
| Nov/Dec/Jan | $-11.7 \pm 1.0$ | 0.94 |

Table 8 Parameters (trend and $R^2$) obtained from the linear regression analysis of tropopause height and TCO for each month of the year, based on data from January 1994 to December 2023. Additionally, results from the Mann-Kendall test are included. Specifically, **Z** represents the Mann-Kendall test statistic, and **S** denotes Sen's slope. **Z** values indicating statistically significant trends at a confidence level of at least 95 % (|**Z**| > 1.96) are highlighted in bold.

| | Tropopause height | | | | TCO | | | |
|---|---|---|---|---|---|---|---|---|
| | Linear regression | | Mann-Kendall test | | Linear regression | | Mann-Kendall test | |
| Month | Trend (m/decade) | $R^2$ | Z | S (m/decade) | Trend (DU/decade) | $R^2$ | Z | S (DU/decade) |
| January | -60 ± 90 | 0.016 | -1.0 | -110 ± 140 | 5 ± 3 | 0.06 | 1.2 | 4 ± 5 |
| February | 70 ± 120 | 0.011 | 0.8 | 100 ± 140 | -2 ± 5 | 0.004 | -0.5 | -4 ± 7 |
| March | 90 ± 80 | 0.04 | 1.0 | 90 ± 130 | 2 ± 3 | 0.018 | 0.6 | 2 ± 5 |
| April | 80 ± 90 | 0.03 | 0.8 | 100 ± 120 | -3 ± 3 | 0.02 | -0.9 | -3 ± 4 |
| May | -30 ± 80 | 0.005 | -0.4 | -20 ± 100 | 1 ± 2 | 0.011 | 1.2 | 3 ± 3 |
| June | 130 ± 60 | 0.14 | **2.2** | 150 ± 80 | -0.4 ± 1.7 | 0.002 | -0.5 | -1 ± 3 |
| July | 120 ± 80 | 0.08 | 1.6 | 200 ± 100 | -0.7 ± 1.6 | 0.008 | -0.5 | -1 ± 2 |
| August | 200 ± 70 | 0.23 | **2.9** | 200 ± 100 | -3.6 ± 1.8 | 0.12 | -1.8 | -4 ± 2 |
| September | 200 ± 100 | 0.14 | **2.1** | 250 ± 150 | -3 ± 2 | 0.05 | -1.4 | -3 ± 3 |
| October | 100 ± 100 | 0.06 | 1.5 | 200 ± 130 | -0.8 ± 1.9 | 0.006 | 0.1 | 0 ± 3 |
| November | 150 ± 90 | 0.09 | 1.9 | 200 ± 100 | 2 ± 2 | 0.03 | 1.0 | 2 ± 3 |
| December | 90 ± 100 | 0.02 | 0.7 | 70 ± 130 | 0 ± 3 | $5 \cdot 10^{-6}$ | -0.2 | 0 ± 4 |

L450: Include reference

This statement certainly requires a reference. The authors have also modified it, changing "contributes" to "may contribute", as ozone depletion depends on several factors, not just on the increase in tropopause height.

With respect to the references, Meng et al. 2021 related the increase in the height of the tropopause to the troposphere warming due to climate change. Furthermore, Match et al. 2022 stated that global warming increases the height of the troposphere, which favours ozone depletion due to chemical processes and reduces its transport into the lower stratosphere.

The modified text will be:

*The increase in tropopause height, primarily related to rising temperatures in the troposphere due to increased concentrations of greenhouse gases (Meng et al. 2021), may contribute to ozone depletion by shifting the ozone layer to higher altitudes (Match et al. 2022).*

---

## Author Comment (AC3)

**Answer to the editor's comment**

Thank you very much for your comment and suggestion. Our response is written in blue in the text.

**Dear authors**

Looking at the reviewer comments and the responses, I have an additional comment.

30 years of AOD time series are a great achievement and to give more valid to the time series I think a figure with the ETC time series at section 2.4.4 would be beneficial.

The figure has been added to the section "ETCs calculation", which is part of Appendix A: Description of selected inputs used in the AOD calculation.

Figure A1 Time series of ETC values for the 320 nm wavelength during 16 intercalibration periods, from 18 August 1993 to 31 May 2024.